# How to make climate-neutral aviation fly

Romain Sacchi [1,5] ✉, Viola Becattini[2,5], Paolo Gabrielli [2], Brian Cox[3], Alois Dirnaichner[4], Christian Bauer [1] & Marco Mazzotti[2] ✉

The European aviation sector must substantially reduce climate impacts to reach net-zero goals. This reduction, however, must not be limited to flight $CO_2$ emissions since such a narrow focus leaves up to 80% of climate impacts unaccounted for. Based on rigorous life-cycle assessment and a time-dependent quantification of non-$CO_2$ climate impacts, here we show that, from a technological standpoint, using electricity-based synthetic jet fuels and compensating climate impacts via direct air carbon capture and storage (DACCS) can enable climate-neutral aviation. However, with a continuous increase in air traffic, synthetic jet fuel produced with electricity from renewables would exert excessive pressure on economic and natural resources. Alternatively, compensating climate impacts of fossil jet fuel via DACCS would require massive $CO_2$ storage volumes and prolong dependence on fossil fuels. Here, we demonstrate that a European climate-neutral aviation will fly if air traffic is reduced to limit the scale of the climate impacts to mitigate.

Today, aviation causes 2.5% of the world's $CO_2$ emissions[1]. Although the last two decades saw a 2% annual improvement in aircraft fuel efficiency[2], $CO_2$ emissions kept growing due to a 4% increase in annual demand, doubling aviation's contribution to global anthropogenic $CO_2$ emissions (barring temporary reductions caused by the Covid-19 pandemic)[3]. Besides flight $CO_2$ emissions, aviation contributes to climate change through "non-$CO_2$ effects" in the atmosphere[4] by releasing short-lived climate forcers (SLCF). Although associated with significant uncertainties, the understanding of non-$CO_2$ effects has improved over the years, allowing characterization of the relationship between atmospheric SLCF emissions and increase in radiative forcing (RF) within acceptable intervals of confidence (see Lee et al.[4] and Allen and co-workers[5–8] for the relevant state-of-the-art).

The European Commission acknowledges the need for policies targeting aviation's full climate impacts[9]; a recent analysis it commissioned[10] suggested ways to regulate non-$CO_2$ effects. Yet they are rarely considered in policy and roadmap documents, misestimating the effort needed to reduce aviation's contribution to climate change. Mitigation relies on improving air traffic management, producing larger and more fuel-efficient aircraft, introducing sustainable aviation fuels, and compensating for any leftover effects. In the roadmap "Destination 2050", EU-based aircraft manufacturers, airports, and airlines aim at cutting $CO_2$ emissions by 92% by 2050 while

compensating the rest through, e.g., carbon removal projects[2]. Other national[11] and international[12–16] organizations, and some airlines[17,18], follow a similar approach. However, these neither include non-$CO_2$ effects nor underline the need for low-carbon electricity for synthetic jet fuels to become climate-neutral[19,20].

Furthermore, a recent report reveals that none of the industry's efficiency or alternative fuel-related targets in the last two decades has ever been met[21]. Finally, the effectiveness of some of the carbon offset schemes may be questioned[22] as, in practice, non-reversibility and avoidance of double-accounting of carbon credits are often not ensured[23,24]. Nevertheless, a recent delegated regulation of the European Parliament and Council establishes a minimum threshold for greenhouse gas emissions savings of synthetic fuels (equal to 70%). It requires such fuels to be produced almost exclusively with additional renewable electricity[25,26].

As the European Union actively develops initiatives to decarbonize aviation, e.g., ReFuelEU[27], we first assess the climate impact of a fossil-based European aviation fleet over the 2018–2100 period under different socio-economic pathways and demand evolution scenarios. Second, we explore the potential of two technology options to limit aviation's climate impact and meet different mitigation scopes: $CO_2$ removal (CDR) and synthetic, electricity-based fuels. Finally, we quantify each technology option's associated life-cycle costs, energy,

[1]Technology Assessment Group, Laboratory for Energy Systems Analysis, Paul Scherrer Institut, Villigen, Switzerland. [2]Institute of Energy and Process Engineering, ETH Zurich, Zurich, Switzerland. [3]INFRAS, Bern, Switzerland. [4]Potsdam Institute for Climate Impact Research, Potsdam, Germany. [5]These authors contributed equally: Romain Sacchi, Viola Becattini. ✉e-mail: romain.sacchi@psi.ch; marco.mazzotti@ipe.mavt.ethz.ch

and natural resources needs. Considering the recent publications on the climate impact of aviation[8,28–32], we provide a comparative overview outlining differences and similarities between our study and the existing literature (Previous works in Methods), highlighting our analysis's completeness and originality. Failing to consider the climate impact caused by the production of synthetic fuels from a life-cycle perspective, as in several recent papers[8,28,30,32], would neglect up to about half of the total impact caused by a growing European fleet. Furthermore, unless demand is reduced, wholly and truly offsetting aviation's climate impact in the future will require an important use of resources, whether synthetic jet fuel is used or not.

## Results

### Climate impact mitigation pathways

Besides the reference scenario where European aviation relies exclusively on fossil jet fuel, we consider two technology options (i.e., mitigation approaches) to reduce aviation's climate impact. First, $CO_2$ removal is performed by Direct Air Capture (DAC) and permanent geological storage of $CO_2$ (aka DACCS) to offset aviation's climate impact for a fossil-based fleet (Fig. 1a). Second, syn-jet fuel (short for synthetic jet fuel) is produced from $CO_2$ captured from the air (i.e., DAC with $CO_2$ utilization, or DACCU) and hydrogen synthesized through water electrolysis, with the remaining impacts offset by DACCS (Fig. 1b). Following the EC-proposed ReFuelEU targets for sustainable aviation fuel (SAF)[27], we assume that syn-jet fuel is initially blended with conventional jet fuel with a volume percentage of 5% in 2030, 63% in 2050, and finally 100% in 2063 (i.e., +2.6% per year). It is an arguably ambitious target because the technology still exhibits a relatively low technology readiness level[33], and scaling up a cost-competitive production might be challenging[19,34]. As we focus on DAC-based approaches, other SAF (e.g., bio- and solar-jet fuels[35]), though worth further investigation, are beyond the scope of this work. While the small quantities of SAF used today are predominantly based on biomass (e.g., produced from used cooking oil)[36], our focus is motivated by the fact that sustainable biomass resources (residues from forestry, agriculture, food) are limited[30,32,37,38,39] and their utilization faces competition. Biomass-based SAF production could be scaled up using dedicated crops. Yet, such feedstock often causes land use

changes and associated climate impacts[40] and further competes with land needed for food production.

To highlight the relevance of non-$CO_2$ effects in designing measures to reduce the climate impact of the European aviation fleet, we consider three scopes of mitigation over the second half of the century: (i) flight-$CO_2$ neutrality, where flight-$CO_2$ emissions only and greenhouse gas (GHG) emissions due to the mitigation approach itself are eliminated (1 *and* 5 in Fig. 1a); (ii) warming neutrality, achieved by mitigating any increase in climate impacts with respect to 2050 levels (1 to 5 in Fig. 1a, and 2 to 5 in Fig. 1b); and (iii) climate-neutrality, achieved by mitigating all climate impacts caused by the fleet from 2018 onwards (1 to 5 in Fig. 1a; 2 to 5 in Fig. 1b). While climate-neutrality implies that the total radiative forcing (RF) caused by the fleet is brought to net-zero after 2050 onwards (with respect to 2018), warming neutrality only requires that the forcing is stabilized at the 2050 level.

Furthermore, we consider three different air-traffic demand trajectories, assuming the European aircraft fleet reaches and exceeds its pre-Covid-19 level by 2024. There is consensus within the industry about this[41], but also the chance that the Covid-19 pandemic may have profound, irrevocable behavior-changing effects on air travel's future demand[42]. The European air-traffic demand trajectories considered from 2025 after the post-Covid recovery onwards are (i) growing demand, whereby the European air traffic, in terms of kilometers flown, converges towards a 1.8% annual growth rate; (ii) stationary demand, whereby European air traffic stabilizes shortly after 2024; and (iii) declining demand, whereby global air traffic reduces at the annual rate needed to achieve warming neutrality without using CDR. Note that we do not consider any potential rebound effects associated with stationary or declining demand[43]−people not spending money on flight tickets might spend it on other goods or services associated with other different climate impacts. However, examining such implications would require different models and approaches that are beyond the scope of this study. Additionally, we have not accounted for any potential shifts in transportation modes. It is however worth noting, that previous research has shown that air travel has by far the most significant climate impacts among all passenger transportation modes[44].

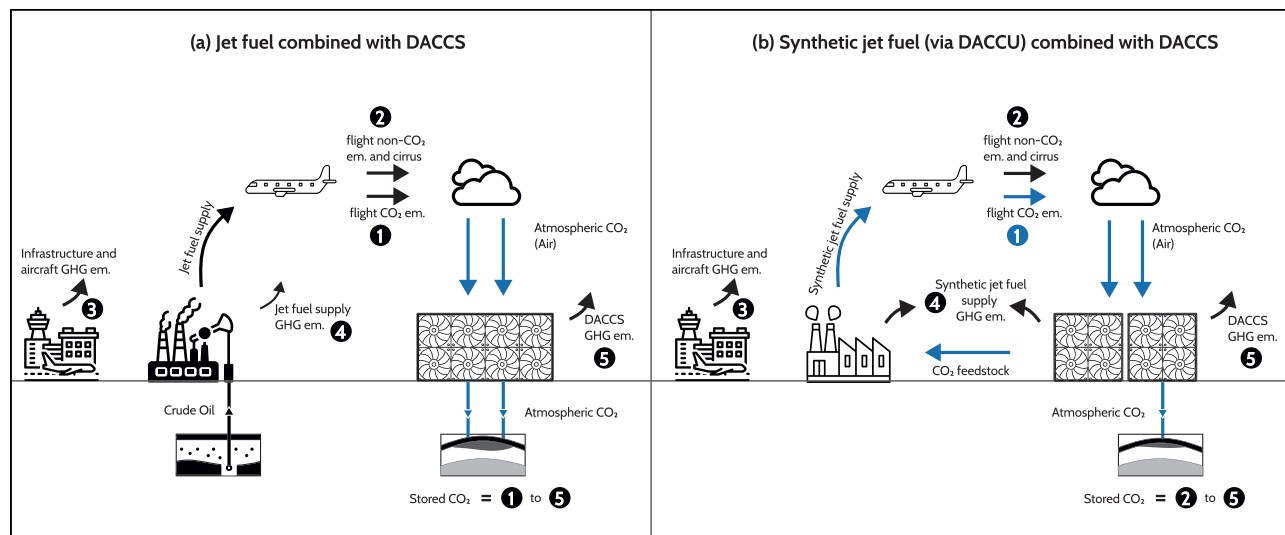

**Fig. 1 | Representation of the technology options considered to reduce the contribution of the European aviation sector to climate change and their associated emission contributions. a** Direct Air Capture (DAC) with permanent geological storage of $CO_2$ to offset the climate change contribution of flight emissions (1 and 2), the supply of infrastructure and aircraft (3), the production and supply of jet fuel (4) and DAC with Carbon storage (DACCS) operations (5); **b** Syn-

jet fuel, produced from electrolysis-based hydrogen and $CO_2$ from DAC via Fischer-Tropsch synthesis, to mitigate flight-$CO_2$ emissions (1), which are considered neutral in this case, and complemented with DACCS to offset the remaining climate change impacts (2 to 5). Note that black and blue arrows represent $CO_2$ flows of fossil and atmospheric origin, respectively.

Furthermore, we consider two future climate scenarios to project future LCA-based GHG emissions embodied in electricity, materials, and services: a baseline scenario in line with the Paris Agreement objectives (i.e., SSP 2-RCP 2.6), aiming at a global temperature increase well below 2 °C by 2100 compared to pre-industrial levels, i.e., the 2 °C scenario; and an alternative scenario (i.e., SSP 2, without a stringent climate mitigation policy), reaching a global temperature increase of approximately 3.5 °C by 2100 compared to pre-industrial levels, i.e., the 3.5 °C scenario. These scenarios offer a wide range of projected cumulative GHG emissions by 2100, which are plausible and consistent with the emissions growth rates of the past two decades[45]. Both climate scenarios consider, to a different extent, the expected improvements in aircraft light-weighting, engine efficiency, seating capacity, and occupancy, but also the progress in other sectors, like electricity and syn-jet fuel production, until 2050. No further advancements are considered after that due to limitations and uncertainty in projected performance, except for the electricity mix, which is projected until 2100 (see Methods for details and values). We exclude other potentially impactful mitigation options (e.g., improved air traffic management, such as navigational avoidance to reroute traffic around ice supersaturated region and mitigate contrail climate forcing[46,47]), revolutionary aircraft designs, biomass-based alternative fuels[48], hydrogen-powered aircraft[49,50], and battery-electric aircraft[50]. In the following, we refer to the 2 °C scenario results, which feature an electricity mix that complies with the recent European delegated regulation for recycled carbon fuels, including synthetic, electricity-based jet fuel[51]. The regulation requires that renewable liquids are produced solely via additional renewable or grid electricity with a greenhouse gas emission intensity below ca. 65 gCO₂/kWh. Results for the 3.5 °C scenario are presented in Supplementary Figs. 1 and 2, reflecting a situation where low-carbon electricity is not available in sufficient amounts.

For each scope of mitigation, air-traffic demand trajectory, and climate scenario, the performance of the two technology options is assessed regarding the emissions of climate forcers, total RF, and CDR requirement for the European fleet. Furthermore, their impacts on resources (costs, electricity, geological $CO_2$ storage capacity, and land and freshwater use) are quantified. To this end, we develop and apply a new life-cycle environmental and cost assessment model for aircraft and electricity-based syn-jet fuel, which is described in the "Methods" section and builds upon earlier work of ours[52,53].

In the following, we distinguish flight emissions (emissions 1 and 2 in Fig. 1) from surface emissions (emissions 3, 4, and 5 in Fig. 1), with the latter caused by the supply of infrastructure, aircraft, fuel (fossil or synthetic), and by DACCS operation. Surface emissions include $CO_2$ and SLCF, such as methane, hydrogen, and refrigerants (CFCs, HCFCs, and HFCs). Flight emissions include $CO_2$ and SLCF, such as Sulphur oxides, black carbon, and water vapor released in the troposphere (i.e., below 9000 m), nitrogen oxides released in the stratosphere (i.e., above 9000 m), and the formation of cirrus clouds. Atmospheric lifetimes and radiative efficiencies of surface forcers are sourced from Chapter 7 of the IPCC AR6 WG1 report[54], while those for flight forcers from Lee et al.[4]. The RF of both types of emissions is calculated via the linear impulse-response model, except for cirrus clouds, where we apply an empirical relationship to correlate the kilometers flown to the cirrus-induced RF, following the literature[4,28]. However, unlike these studies, which use the so-called GWP* metric introduced by Allen and colleagues[5,6,55,56], the warming contribution of flight and surface emissions is expressed as time-series of $CO_2$ emissions that would cause identical warming through the Linear-Warming-Equivalent (LWE) method. The LWE method is introduced by Allen et al.[7] and is both exact and metric independent. The amount of $CO_2$-LWE to sequester to compensate for the warming caused by SLCF emissions is calculated by inverting the linear impulse-response model routinely used for metrics calculation. We present this approach in detail in Radiative

forcing and warming contribution of emissions in Methods. Another significant difference with the method used in Brazzola et al.[28] is that we consider the life-cycle emissions of $CO_2$ and SLCF related to the manufacture of the aircraft, infrastructure (e.g., airport), fuel (including electricity), as well as to DACCS operation. Applying the LWE method to a time-series of emissions calculated by prospective life-cycle assessment is an original approach.

## Conventional jet fuel vs. synthetic jet fuel

We analyze two technology options: the European aviation fleet relies on fossil-based jet fuel (Fig. 2) or syn-jet fuel with a blend volume percentage increasing from 5% in 2030 to 63% in 2050 and 100% by 2063 (Fig. 3). In both cases, DACCS is deployed to mitigate the remaining contributions to climate change. The syn-jet fuel is produced through Fischer-Tropsch synthesis, fed by hydrogen ($H_2$) from water electrolysis and carbon monoxide (CO) from the reverse water-gas shift reaction using $CO_2$ from DAC at generic European locations—further information on the relevant processes is given in Methods.

Growing air-traffic demand. Despite higher fuel efficiency and larger seating capacity and occupancy than today, a fossil-based fleet growing unmitigated at its average historical rate will directly emit 24 Gt $CO_2$ during the 2018–2100 period (Fig. 2a), while SLCF would cause over two-thirds of the RF (Fig. 2d). By 2100, unmitigated emissions of $CO_2$ will increase three-fold relative to 2018 (Fig. 2d). Using low-aromatic, hydrogen-rich syn-jet fuel appears attractive as it avoids flight-$CO_2$ emissions of fossil origin (Fig. 3a) while reducing soot and ice particle formation at high altitudes in ice supersaturated regions (based on data extrapolation from ref. 57). This reduces the cloudiness and lifetime of cirrus clouds and their associated effective RF by 65%[58]. Overall, using 100% syn-jet fuel reduces the fleet-induced climate warming over the 2018–2100 period, as opposed to using jet fuel, with a Global Mean Surface Temperature (GMST) increase of +0.02 °C instead of +0.035 °C (Fig. 2d vs. Fig. 3d). In addition to stabilizing the contribution of flight-$CO_2$ to the total RF after 2063, the increase of syn-jet fuel share in the blend decreases flight SLCF emissions (Fig. 3a). Unfortunately, this positive effect is counterbalanced by the increase in fleet activity, resulting in a growing share of RF due to SLCF emissions (Fig. 3d). Furthermore, the forcing caused by surface $CO_2$ emissions increases linearly until 2100 (Fig. 3d). As the annual production of syn-jet fuel increases from 108 billion liters in 2063 to 202 billion liters in 2100, $CO_2$ emissions associated with DAC and $H_2$ production increase, despite the drop in the carbon intensity of the electricity mix (i.e., from 400 g $CO_2$/kWh today to 27 g and 20 g in 2050 and 2100, respectively). Furthermore, $H_2$ leaks, represented by a 1% mass loss related to venting, storage, and boil-off along the supply chain[59], also contribute to the overall forcing induced by a syn-jet fuel-based aviation by extending the atmospheric lifetime of methane[60] (Fig. 3d, Surface-others).

Should mitigation measures be deployed, the amount of $CO_2$ to sequester via DACCS would depend on the mitigation scope and be lower for the fleet relying on syn-jet fuel (Figs. 2g and 3g). Regardless of the fuel, achieving climate-neutrality from 2050 onwards implies an important removal in the first year (2050) to offset the cumulated RF caused by aviation between 2018 and 2049. This is difficult in practice, and CDR could start well before 2050 to accommodate a more feasible trajectory of emissions reduction. It is followed by an increasing removal effort due to the rising RF induced by the fleet. Pursuing warming neutrality would reduce the CDR effort needed over the 2050–2100 period by 40% (syn-jet fuel) to 50% (jet fuel), stabilizing the GMST increase caused by the fleet to ca. +0.012–0.016 °C (Figs. 2d and 3d).

Most importantly, failing to consider the additional RF caused by the production of syn-jet fuel from a life-cycle perspective, as in Brazzola et al.[28], would neglect more than a third of the impact caused by a growing European fleet activity (Fig. 3d), despite a very ambitious

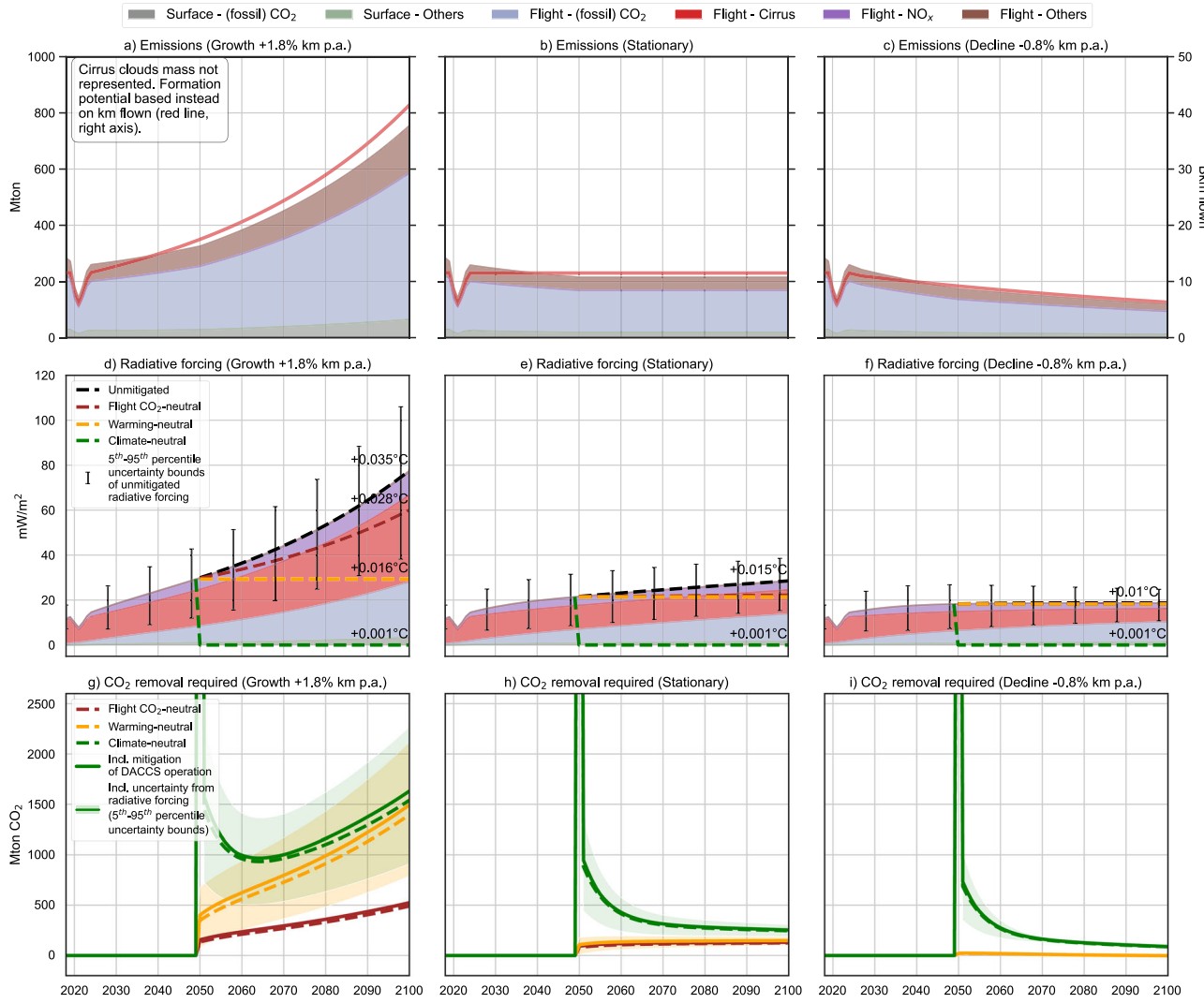

**Fig. 2 | European aviation fleet activity for the 2018–2100 period relying on jet fuel and carbon dioxide removal (CDR) performed via direct air capture and carbon storage (DACCS) to meet the mitigation scope, for the three air-traffic demand trajectories, under the 2 °C climate scenario.** Panels **a**, **b**, **c** amount of climate forcers emitted (left *y*-axis) and of kilometers flown (right *y*-axis); Panels **d**, **e**, **f** Radiative forcing (RF) of climate forcers and forcing trajectories for the mitigation scopes considered (note: the additional RF caused by DACCS is calculated iteratively and included in the total RF). The Global Mean Surface Temperature (GMST) increase by 2100 relative to 2018 is indicated in degrees Celsius; Panels **g**, **h**, **i** CDR requirement (with and without additional removal needed to mitigate DACCS operations) to meet a given mitigation scope over the 2050–2100 period. Error bars represent the uncertainty around the radiative efficiency of flight Short-lived Climate Forcers (SLCF) emissions.

climate scenario in which fossil fuels in the European power sector are phased out already by 2030. Mitigating solely flight-$CO_2$ emissions would require a significantly smaller CDR effort. Still, it would reduce the GMST increase by 20% only with respect to the unmitigated scenario (+0.028 °C vs. +0.035 °C, Fig. 2d), thus leaving up to 80% of the climate change impacts unmitigated. Additionally, as the share of syn-jet fuel in the blend increases, the decrease in the RF caused by less persistent cirrus clouds (based on refs. 57,58) counterbalances the warming caused by other forcers, resulting in a net negative amount of CDR (Fig. 3g).

The GHG emissions associated with electricity supply for DACCS operation increase the overall CDR requirement by 13% in 2050, but this decreases to 5% by 2100 as the electricity decarbonizes (see the difference between dashed and solid lines in Figs. 2g and 3g).

The uncertainty around the RF caused by non-$CO_2$ effects is significant. The uncertainty is discussed by Lee et al.[4] and represented in Figs. 2 and 3 by the error bars obtained from the 5th and 95th percentile of the RF indices for each non-$CO_2$ effect—also from ref. 4. Most of the spread stems from the uncertain time- and location-dependent

impact of cirrus cloud formation. In this respect, the more recent work of Digby et al.[61] indicates a lower central estimate for the RF of cirrus cloud formation while maintaining equally important uncertainty ranges. Consequently, climate- and warming-neutrality exhibit wider uncertainties than flight $CO_2$-neutrality: the CDR requirement scales with the uncertainty associated with non-$CO_2$ effects, which are left unmitigated in the case of flight $CO_2$-neutrality, as shown by the shaded areas in Figs. 2g and 3g.

**Stationary air-traffic demand.** Stabilizing the fleet activity at the 2024 level results, in the second half of the century, in a constant forcing caused by SLCF (mostly cirrus and $NO_x$) and an increasing forcing from $CO_2$ (either flight or surface $CO_2$) because $CO_2$ cumulates (Figs. 2e and 3e). The climate impacts of the fossil- and syn-jet fuel-powered European aviation fleets will decrease in 2100 by 57% and 55%, respectively, with respect to the growth scenario.

As the net contribution of aviation to climate change reduces, the need for compensation via DACCS to achieve climate-neutrality decreases for both fuel options (Figs. 2h and 3h). Considering a fossil-based fleet, flight-$CO_2$ and warming-neutrality mitigation targets

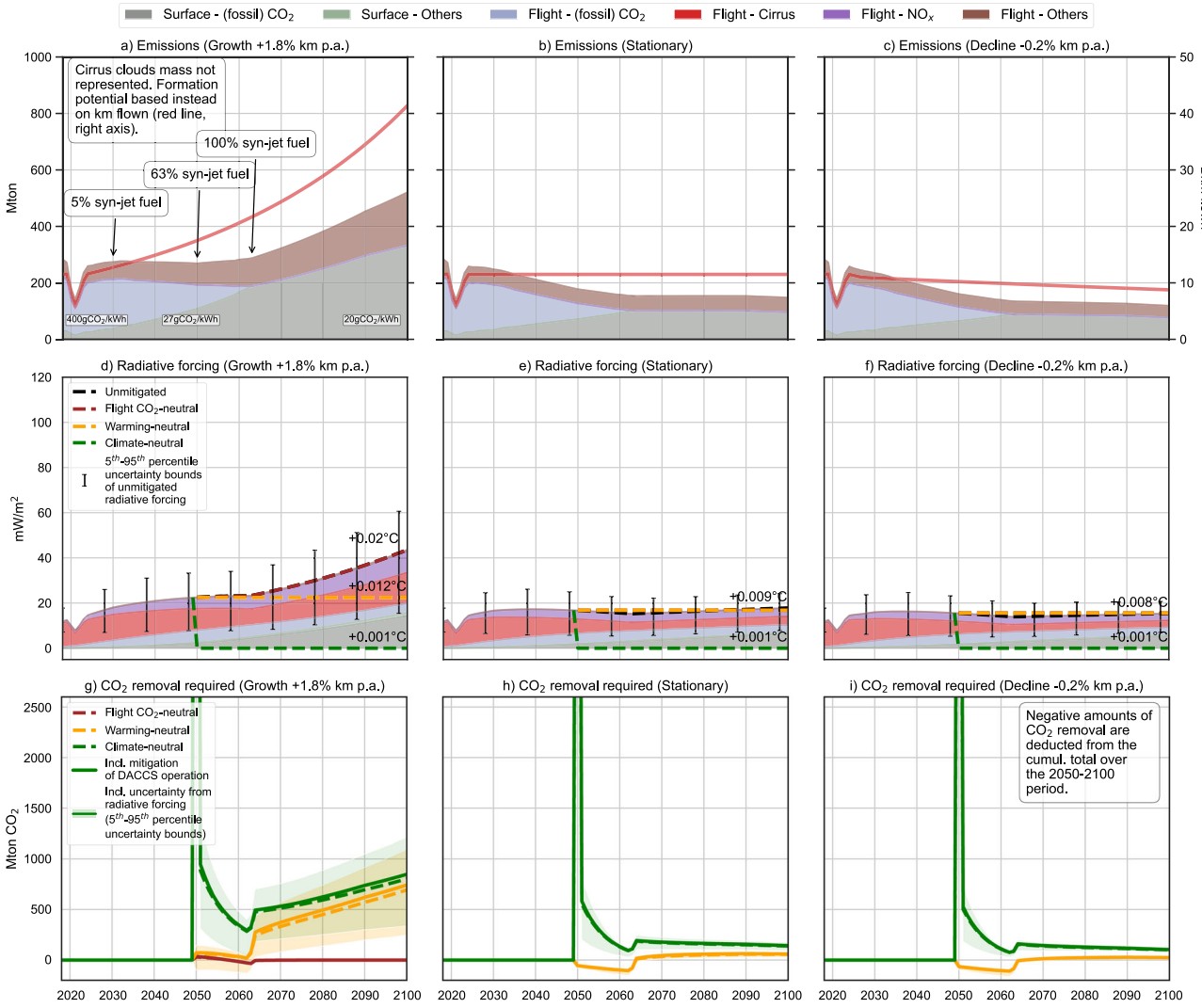

**Fig. 3 | European aviation fleet activity for 2018–2100 relying on syn-jet fuel and carbon dioxide removal (CDR) performed via direct air capture and carbon storage (DACCS) to meet the mitigation scope, for the three air-traffic demand trajectories, under the 2 °C climate scenario.** Panels **a**, **b**, **c** amount of climate forcers emitted (left y-axis) and kilometers flown (right y-axis). The $CO_2$ intensity of electricity along the time axis is indicated at the bottom and is provided by the climate scenario; Panels **d**, **e**, **f** Radiative forcing (RF) of climate forcers and forcing trajectories for the mitigation scopes considered (note: the additional RF caused by DACCS is calculated iteratively and included in the total RF). The global mean surface temperature (GMST) increase by 2100 relative to 2018 is indicated in degrees Celsius; Panels **g**, **h**, and **i** CDR requirement (with and without additional removal needed to mitigate DACCS operations) to meet a given mitigation scope over the 2050–2100 period. Error bars represent the uncertainty around the radiative efficiency of flight short-lived climate forcers (SLCF) emissions.

tend to converge as the forcing caused by SLCF is kept at the 2024 level and thus requires a similar amount of CDR (Fig. 2h). Using syn-jet fuel, *warming neutrality* will be achieved as soon as 2050 without DACCS, thanks to the stabilization of the fleet activity (Fig. 3e). Indeed, the RF caused by the European fleet decreases with respect to 2050, resulting in a negative CDR requirement if warming neutrality is pursued. However, as the fleet relies exclusively on syn-jet fuel from 2063 onwards, the cooling effect of the decreasing RF from SLCF is counterbalanced by the increase in surface $CO_2$ emissions due to fuel production (primarily hydrogen synthesis). Similarly, flight-$CO_2$ neutrality is achieved after 2050 using syn-jet fuel without DACCS, owing to the constant fleet activity after this onset year (Fig. 3e).

Declining air-traffic demand. A limited decline in aviation activity (i.e., kilometers flown) of up to 0.8% per year if the fleet is powered by jet fuel, would eliminate the need for CDR to reach flight-$CO_2$ and warming-neutrality (Figs. 2i and 3i). Such decline could be limited to 0.2% annually if the fleet uses syn-jet fuel instead. The decrease in SLCF emissions would counteract the additional RF caused by the accumulation of $CO_2$ with respect to 2050 (Figs. 2f and 3f). The cooling

effect (relative to the onset year) caused by the fall in SLCF emissions would effectively act as an offset mechanism for $CO_2$ emissions. For warming- and flight-$CO_2$ neutrality, DACCS need is virtually eliminated thanks to the continued decrease of SLCF emissions (mostly reduced formation of cirrus clouds). On the other hand, climate-neutrality cannot be achieved through demand-reduction measures alone. It requires a large amount of CDR in the first year (2050) to compensate for the RF cumulated between 2018 and 2049, followed by a decreasing CDR rate to compensate for newly emitted forcers, to an extent similar to the Stationary scenario (Figs. 2i and 3i).

### Climate-neutral European aviation and resources
While syn-jet fuel and DACCS could offer feasible climate impact mitigation pathways for the aviation sector, Fig. 4 puts these options in the context of resources needed.

If growth in air-traffic demand is sustained, using syn-jet fuel combined with DACCS would require substantial resources, among which deeply decarbonized electricity (i.e., with a carbon intensity of 20 $gCO_2$/kWh in 2100). To achieve climate-neutrality, almost 70 times

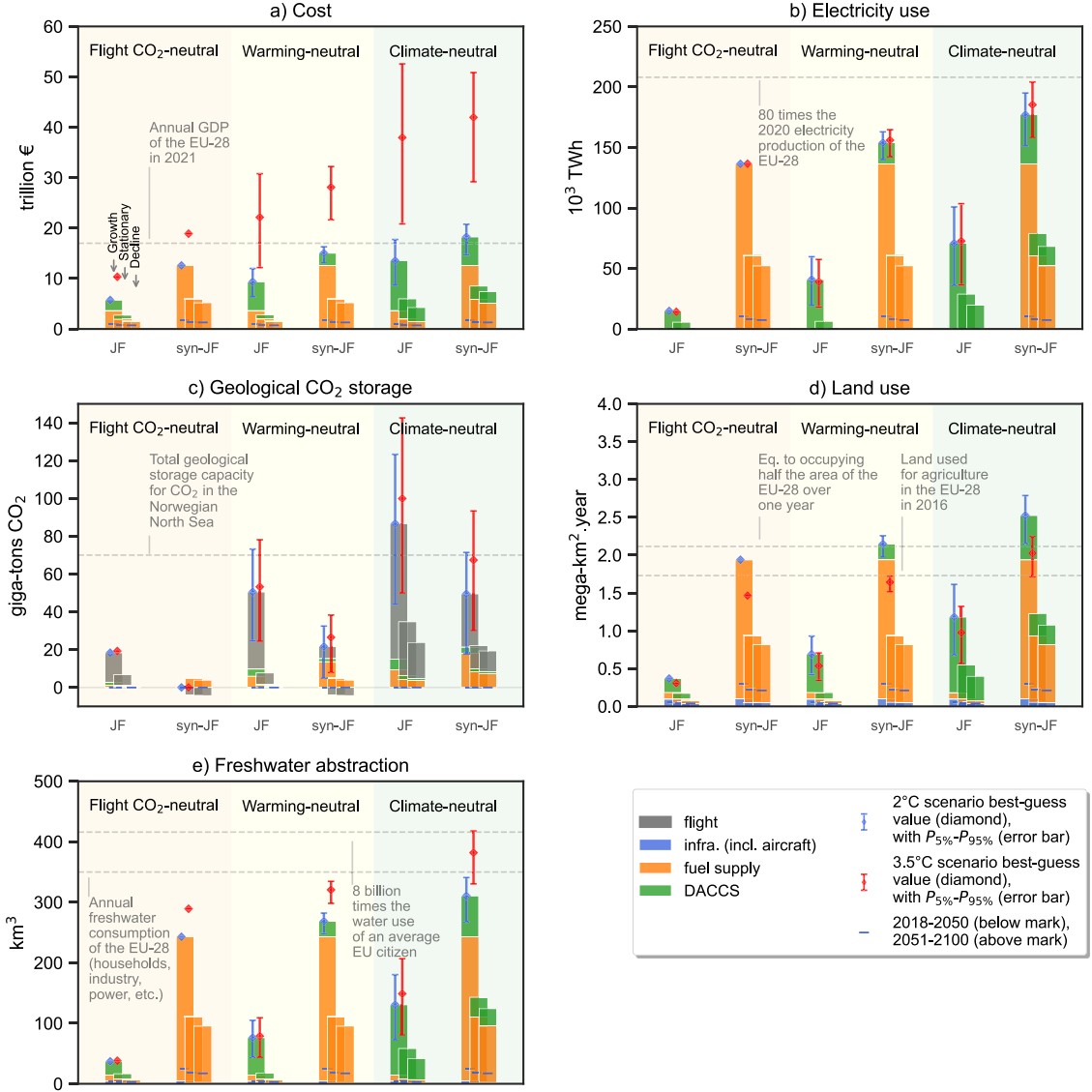

**Fig. 4 | Cumulative resources required to achieve flight-CO₂, warming, and climate-neutrality for a growing, stationary, and decreasing European passengers' fleet activity over the 2018–2100 period, using jet fuel and syn-jet fuel complemented by direct air capture and carbon storage (DACCS), considering a 2 °C (stacked bars and blue diamonds) and 3.5 °C climate scenario (red diamonds). a** Cumulative cost, in trillion Euros, in today's terms. **b** Cumulative electricity use, in thousands of terawatt-hours. **c** Cumulative geological storage, in billions of tons of CO₂ stored. **d** Cumulative land use occupation, in millions of square kilometers-year. Land use is expressed as the area used over one year. **e** Cumulative freshwater abstraction, in cubic kilometers. Freshwater abstraction considers freshwater uptake but leaves out evaporation or release. Error bars correspond to the 90% confidence interval quantifying the non-CO₂ effects. JF (fossil) jet fuel, syn-JF syn-jet fuel.

the electricity production of the European Union in 2020[62] (i.e., 2600 TWh) would be needed cumulatively between 2018 and 2100 (~180,000 TWh)—but mainly during the second half of this century, and predominantly for hydrogen production. In other words, 1.3 times the current annual electricity output in the EU-28 would be needed each year between 2050 and 2100. The cumulative abstraction of freshwater and land use between 2018 and 2100 would also reach excessive levels: 200 to 250 million hectares-year of land would be required, while the freshwater needed would correspond almost to the annual consumption of the EU-28. Most of the use of land and freshwater stems from renewable electricity production (i.e., solar, wind, biomass, and hydropower plants), despite considering efficiency improvements in the LCA database (e.g., the area occupied by PV panels per kW installed decreases by 50% between 2020 and 2050 to reflect the expected increase in the conversion efficiency); such footprints could be partially reduced by using an alternative low-carbon

electricity mix, but at the risk of impacting other resources—see *Sensitivity analyses* in the Supplementary Information file. Such massive demand for decarbonized electricity, land, and freshwater suggests that Europe's entire large-scale syn-jet fuel supply may not be produced domestically. At the same time, other regions like the Middle East, Australia, or South America might offer more promising opportunities to supply the required resources (especially land and renewable electricity)—provided they satisfy their own domestic demand. Resource requirements for syn-jet fuel production might also be somewhat lower in those regions due to higher renewable yields than in Europe. On the other hand, using (fossil) jet fuel and offsetting the climate impacts via DACCS would decrease the use of electricity, land, and freshwater by 50 to 60%, relative to using syn-jet fuel complemented with DACCS. However, this would prolong our dependence on fossil energy and require a CO₂ storage capacity larger than the proven storage capacity of the Norwegian continental shelf. This CO₂

storage capacity may be a limited resource due to both limitations in scale-up and competition for $CO_2$ storage space with other hard-to-decarbonize sectors[63]. On top of the potential resource demands we have quantified, there can be further constraints. A massive expansion of renewable power generation, hydrogen production, and electric mobility—all critical to net-zero $CO_2$ emissions—are associated with substantial increases in demand for certain materials, which might exceed production capacities[64–67].

Under the 3.5 °C scenario, land occupation decreases compared to the 2 °C scenario due to lower shares of renewable power. In contrast, the abstraction of freshwater increases (i.e., among other reasons, more cooling water is used as the share of combustion-based power plants is more prominent with respect to the 2 °C scenario). Moreover, the $CO_2$ storage requirements are higher, as larger amounts of CDR are needed to mitigate the emissions caused by using less clean electricity for fuel production and operation of DACCS. Finally, using (fossil) jet fuel rather than syn-jet fuel and offsetting the climate change contribution with DACCS is less costly across mitigation scopes (despite the cost reductions due to learning by doing). However, as the mitigation scope becomes more stringent (i.e., warming-neutrality), the differences in costs between the two fuel options disappear. The climate scenario seems to be a stronger cost determinant than the fuel option. Reducing the demand for air traffic should be a priority as it significantly decreases the resources needed, regardless of the climate scenario or technology option. Despite uncertainties and assumptions taken, the outcomes of our analysis are robust, as an extensive sensitivity analysis in Supplementary Fig. 4 shows. There, we test the sensitivity of the results to variations of several key input parameters such as cost and origin of electricity, radiative forcing of cirrus clouds, annual fleet growth rate, the efficiency of aircraft, and blending targets for syn-jet fuel.

## Discussion

This study quantifies the efforts needed to mitigate the contribution of the European aviation fleet to global warming through an offsetting approach based on $CO_2$ removal by Direct Air Capture and permanent geological storage of $CO_2$ (i.e., CDR) and the adoption of synthetic, electricity-based jet fuel produced from $CO_2$ captured from air and hydrogen synthesized through water electrolysis. Over the 2018–2100 period, our results agree on two points with the recent findings of Grewe et al.[29] and Klöwer et al.[8]. First, given their significance, non-$CO_2$ effects—particularly those triggered by $NO_x$ emissions and cirrus clouds formation—should be considered when laying out plans to reduce aviation's contribution to climate change. It is, however, important to acknowledge the significant uncertainty surrounding their RF potential. For example, the recent work of Digby et al.[61] suggests that the warming effect of cirrus clouds may be lower than the estimate used in this study—which we quantify as part of a sensitivity analysis in the Supplementary Information. Second, the non-$CO_2$-to-$CO_2$ warming effect ratio increases as the aviation sector grows.

Furthermore, our results also agree with Brazzola et al.[28]: there is some ambiguity around the notion of "net-zero" put forward by the different roadmaps and agendas. This ambiguity may be misleading: "net-zero" regarding flight $CO_2$ emissions only superficially tackles the GMST increase caused by the fleet. In contrast, "net-zero" regarding climate impacts implies deploying resources to an extent not fully understood until now.

Generally, the extent to which CDR and resources are needed depends on three factors.

First and foremost, the mitigation scope: simply offsetting flight-$CO_2$ emissions requires limited or no CDR (up to 560 Mt $CO_2$ yearly in 2100, under a growing demand trajectory) and may be achieved by adopting syn-jet fuel produced from $CO_2$ captured from the air. Yet, unless this fuel is made with fully decarbonized electricity, such a mitigation scope appears ill-defined as it would simply move $CO_2$

emissions upstream to the fuel production stage. And most importantly, this would only avoid ~20% of the GMST increase over 2018–2100. Achieving warming- or climate-neutrality, that is, off-setting any additional contribution to global warming relative to an offset year (i.e., 2050 in our analysis) or offsetting any past and future contribution to global warming, would require significant use of CDR. For climate-neutrality, 1 and 1.7 Gt $CO_2$ must be removed annually between 2050 and 2100 for the syn- and (fossil) jet fuel options under a growing demand trajectory in a 2 °C climate scenario, respectively (and 20% more in a 3.5 °C climate scenario). On the other hand, this would reduce the GMST increase over 2018–2100 by more than 95%. It is worth noting that, compared to the overall CDR requirements in the ensemble of IPCC's 2 °C scenarios, in which between close to zero and up to 5 Gt $CO_2$ in 2050 and about 5-15 Gt $CO_2$ in 2100 need to be removed[68], the amounts needed for a climate-neutral European aviation under the growing demand trajectory are substantial.

Second, the technology option deployed matters: using syn-jet fuel instead of fossil jet fuel can result in a lower requirement for $CO_2$ storage (up to 45% lower to achieve climate-neutrality, under a growing demand trajectory), but only if produced with low-carbon electricity. Hence, using syn-jet fuel depends on adding low-carbon power generation capacities.

Third, the CDR and resource requirements depend on the air-traffic demand trend and the contribution of non-$CO_2$ effects: mitigating the climate impact of the European aviation fleet with DACCS will be highly resource intensive unless demand-reduction measures are adopted. Such actions would decrease non-$CO_2$ effects (i.e., a cooling effect relative to the current warming induced by aviation) that compensates for the continued accumulation of aviation $CO_2$ emissions to an extent where CDR may not even be required to achieve flight-$CO_2$ or warming neutrality. However, this only holds if demand decreases. Once a new, albeit reduced, stationary demand level is attained, the cooling effect caused by the downward emission trend of SLCF (i.e., non-$CO_2$ effects) wears off, and CDR is required again to compensate for the warming caused by $CO_2$ emissions.

In summary, if aviation growth is sustained, fully mitigating the climate impacts caused by the European aviation sector this coming century through offsetting and the adoption of syn-jet fuel will simultaneously require CDR and significant amounts of energy, natural and financial resources—despite (i) avoiding flight-$CO_2$ emissions of fossil origin by using synthetic jet fuel, (ii) technical and economic improvements in aircraft and fuel production, (iii) decarbonized energy supply and (iv) considering lower bound levels for non-$CO_2$ effects. Thus, from a physical standpoint, reducing air-traffic demand is a good short- to mid-term solution. It drastically reduces the scale of the environmental and economic effort needed to limit the impact of aviation on the climate. Doing so gives society time to develop other, possibly longer-term, sustainable solutions (e.g., navigational avoidance, hydrogen-powered and battery-electric aircraft, and other CDR options), which may be combined with the ones addressed in our work.

## Methods
### General workflow
The European air-traffic demand trajectories are modeled between 2018 and 2100, based on and extrapolated from the Destination 2050 report projections for demand[2] in the case of the growth trajectory, to derive the flown distance and amount of passengers to transport over the period. Considering this, an iterative spreadsheet-based model that combines life-cycle energy and material inventories and costs (available as part of the Supplementary Information material) is used to dimension the aircraft of the European fleet. It calculates the emissions during take-off, climb, cruise, descent, and landing, fuel supply, infrastructure, aircraft production requirements, and associated emissions and resource impacts. The RF contribution of surface

and flight emissions are calculated for the three different demand trajectories, two fuel options, and two socio-economic pathways for the future development of the economy ("climate scenarios"). The requirement in terms of CDR (i.e., DACCS in this case) is calculated iteratively to attain the mitigation scope and mitigate the emissions of its operation.

## Previous works

Table 1 contains a summary of key aspects in the evaluation of climate impacts of aviation and the way these were considered in various recent studies on this topic. This brief overview shows that they partially exhibit substantial simplifications and shortcomings, which could limit the reliability of the outcomes of some of the analyses. It also highlights the uniqueness and completeness of our assessment.

## Life-cycle environmental and cost modeling

The life-cycle model in the spreadsheet supporting this study is at the center of the framework used to produce results. The model calculates the required material, energy flows, and related emissions directly and indirectly needed to support a flight's different life-cycle stages. The life-cycle steps considered include the aircraft manufacture and maintenance, the construction of the airport, the production and distribution of the fuel (i.e., conventional jet fuel and syn-jet fuel), the operation of the mitigation chain (DAC and CCS) as well as the aircraft operation itself (i.e., take-off, climb, cruise, descent, and landing). Note that the End-of-Life processing of the aircraft is not considered. The model derives material, energy, resource, emissions, and cost indicators based on a set of parameters: the flight distance, the fuel blend, the aircraft type, the year of manufacture (and operation), the occupancy as well as the climate scenario considered. The results are further normalized by the flight distance and the passenger occupancy to derive indicators on a passenger-kilometer basis, then multiplied by the overall demand expressed in passenger-kilometers. Such an approach follows the principles of the ISO 14040 standard series[69]. It ensures a standard accounting of material and energy use and related emissions that support the realization of a functional unit (i.e., the transport of a passenger over one kilometer).

The life-cycle model sizes the aircraft based on the flight distance and year of construction (which conditions improvement factors to apply). Details are given in Aircraft. The approach used to calculate in-flight emissions is described in Surface and flight emissions. The approach used to calculate the RF attributed to each surface and flight emission is presented in Radiative forcing and warming contribution of emissions. The emission rate of these emissions in the atmosphere depends on the fleet demand trajectory considered, which modeling is explained in European fleet scenarios. The approach used to calculate direct operating costs is presented in Costs. The remaining processes that fall outside the immediate scope of the aircraft life-cycle are modeled using a life-cycle inventory database. This database is adapted to the aircraft production, operation year, and climate scenario. The approach used to adjust the life-cycle inventory database to a specific socio-economic-climate and temporal context is explained in Prospective life-cycle inventory database.

Life-cycle GHG emissions, costs, and resource indicators are calculated for representative flights by broad destination departing from Europe (i.e., North America, South America, Africa, Middle East, Asia, and intra-European flights). The destination type presets the altitude profile of the flight, notably the share of the fuel emissions released in the different strata of the atmosphere—see Air emissions. Longer flights tend to spend a higher percentage of the flight distance at high altitudes than shorter flights. On an average Europe-Asia journey, 97% of the route is flown above 9000 m of altitude, against 85% for an average intra-European flight. This is important as the different non-$CO_2$ effects in this study occur in different altitude ranges, while $CO_2$ emissions have a constant and similar impact on warming regardless of

the emission altitude. The results per passenger-kilometer are multiplied by the demand for transport per destination in each air traffic demand case to obtain overall indicators for the European fleet.

## European fleet scenarios and demand projection

European air-traffic scenarios are calculated for each demand trajectory, starting from 2018. European air traffic is comprised of flights departing from the EU-27 as well as EFTA member states. The total transport demand in passenger-km by the year 2050 is estimated for the growing demand trajectory based on projections from the "Destination 2050" roadmap report[2]. The report indicates an expected annual growth in passengers of 2% until 2050: a 1.4% yearly increase in flights, combined with an increase in seating capacity and a load factor of 0.3%. We further assume a similar development over the 2050–2100 period. The roadmap report "Destination 2050" also shows demand per destination region. Assuming an average distance for each destination with Paris (FR) as a departure point, we derive the overall transport demand in passenger-kilometers based on the traffic split by destination. For the average distance of intra-EU flights, we refer to ref. 70. Hence, traveling within the European Union (Intra-EU) spans a distance of about 1,000 kilometers. If one were to embark on a journey from Paris, France, to different regions of the world, the distances would vary. A trip to North America, using Chicago, Illinois, USA as a destination, would cover approximately 6654 kilometers. Heading to South America, considering Manaus, Brazil as a destination, from Paris would be about 8306 kilometers. If Africa is the destination, considering Bangui in Central Africa as a destination, the distance from Paris would be around 5012 kilometers. Travel to the Middle East, using Iran as a reference point, would involve a journey of about 4538 kilometers. Lastly, a trip from Paris to Asia, with Beijing, China as the endpoint, would span approximately 8216 kilometers.

An annual growth rate of zero is used for the stationary demand trajectory. Finally, the "negative" growth rate for the declining demand trajectory is defined as the minimum decline that allows avoiding the use of DACCS to reach warming-neutrality. This depends on the climate scenario and fuel technology considered. Note that we preserve the traffic split by destination in the stationary and declining demand trajectories. Details on how the air-traffic demand is broken by destination are provided under the "European fleet" tab in the spreadsheet model and plotted in Fig. S3 in the Supplementary Information.

## Aircraft

Flights departing from Europe to each destination are modeled using the generic average-performing aircraft and flight altitude profiles described in Table 2. The aircraft model primarily builds on and extends the work conducted by Cox et al.[52]. Improvements in weight and aerodynamics are taken from Cox et al.[52], sourcing from various projections[71–73]. They are described in detail under the "Aircraft specs" tab of the spreadsheet model. The fuel consumption of the different aircraft is calculated separately for take-off, climb, cruise, and descent and appears to be quasi-linearly related to the mass, based on refs. 52,74. Expected annual improvements in terms of fuel efficiency, seating capacity, and load factor between today and 2050 are aligned with those from the Destination 2050 roadmap report[2]: 0.6%, 0.3%, and 0.3%, respectively, to reach a cumulative improvement between 2020 and 2050 of 20%, 9% and 9%. These improvements correspond to a yearly decrease of the fuel burn rate (in liters of jet fuel per passenger per km) of 1.2% between 2018 and 2050.

Further specifications on aircraft used for modeling the European fleet activity are available under the Scenarios tab in the spreadsheet model. We do not consider other improvements regarding light-weighting, occupancy rate, and fuel consumption after 2050. Also, we do not consider the End-of-Life treatment of the aircraft, as this life-cycle phase is known to have negligible impacts[52]. Finally, it is relevant to mention that we do not consider the weighted age of the fleet but

**Table 1 | Key aspects regarding the evaluation of climate impacts of aviation and their consideration in recent publications. "SAF" refers to drop-in hydrocarbon produced from hydrogen and CO$_2$ via Fischer-Tropsch synthesis**

| Study | Net carbon intensity of SAF (including combustion and excluding non-CO$_2$ climate impacts due to combustion) | Approach for quantification of equivalence between CO$_2$ and non-CO$_2$ climate impacts due to short-lived climate forcers | SAF non-CO$_2$ climate impacts | Consideration of radiative forcing beyond direct flight CO$_2$ emissions and due to short-lived climate forcers | Consideration of radiative forcing of H$_2$ emissions |
|---|---|---|---|---|---|
| This study | Quantified using a Life Cycle Assessment (LCA) approach. | Linear-Warming-Equivalent (LWE) method. | Modeled according to syn-jet fuel properties (which leads to a reduction of radiative forcing). | Included; considering all short-lived greenhouse gases according to IPCC AR6[54] using an LCA approach. | Included; according to ref. 60 |
| Brazzola et al.[28] | Assumed to be zero. | GWP*. | Modeled according to syn-jet fuel properties (which leads to a reduction of radiative forcing). | Not included; lack of a complete LCA approach. | Not included. |
| Klöwer et al.[8] | Assumed to be zero. | Linear-Warming-Equivalent (LWE) method. | Only reduced contrail formation due to lower soot emissions considered. | Not included; lack of a complete LCA approach. | Not included. |
| Grewe et al.[29] | Generic assumption: Minus 65% net CO$_2$ emission intensity compared to fossil kerosene in 2020, increasing to minus 80% in 2100. | Using the AirClim model[106,107], the authors do not establish an equivalence between CO$_2$ and SLCF. They calculate the effect on the climate of the various forcers using the simplified linear model *AirClim*. However, in Appendix B, they establish a simplified relationship between the forcing of CO$_2$ and the forcing of O$_3$—a simplified LWE method | Only reduced contrail formation due to lower soot emissions considered. | Not included; lack of a complete LCA approach. | Not included. |
| Bergero et al.[30] | Assumed to be zero. | GWP20/50/100, GTP20/50/100. | Assumed to be equal to those of fossil kerosene. | Not included; lack of a complete LCA approach. | Not included. |
| Planès et al.[31] | Generic assumption: Minus 75% net CO$_2$ emission intensity. | GWP*. | Assumed to be equal to those of fossil kerosene. | Not included; lack of a complete LCA approach. | Not included. |
| Dray et al.[32] | Assumed to be zero. | GWP20/100/500. | Modeled according to syn-jet fuel properties (which leads to a reduction of radiative forcing). | Partially included: CO$_2$, CH$_4$, and N$_2$O related climate impacts from fuel supply; lack of a complete LCA approach. | Not included. |

**Table 2 | General European fleet aircraft specifications and altitude profiles per destination type**

| Flight to | Aircraft type | Lifetime | Flight length | Operating weight | | | Seating capacity | | | Passenger-eq. occupancy[a] | | | Fuel consumption | | | Share of fuel burned above 9000 m[77] |
|---|---|---|---|---|---|---|---|---|---|---|---|---|---|---|---|---|
| | | Million km | km | kg | | | Unit | | | Unit | | | Liter kerosene per occupied seat per 100 km | | | % |
| | | | | 2020 | 2030 | 2050 | 2020 | 2030 | 2050 | 2020 | 2030 | 2050 | 2020 | 2030 | 2050 | |
| Intra-Europe | Large, narrow body | 50 | 1,000 | 48,200 | 47,800 | 49,000 | 115 | 118 | 126 | 94 | 100 | 113 | 3.8 | 3.3 | 2.6 | 72% |
| North America | Small, wide body | 90 | 6,654 | 121,000 | 117,000 | 112,000 | 211 | 218 | 231 | 179 | 190 | 214 | 3.2 | 2.8 | 2.1 | 96% |
| South America | Long, wide body | 115 | 8,306 | 187,800 | 180,500 | 170,700 | 305 | 314 | 333 | 251 | 266 | 300 | 3.4 | 2.9 | 2.1 | 97% |
| Africa | Small, wide body | 90 | 5,012 | 93,600 | 90,900 | 87,200 | 178 | 183 | 194 | 146 | 155 | 175 | 3.2 | 2.8 | 2.1 | 95% |
| Middle East | Small, wide body | 90 | 4,538 | 65,300 | 63,600 | 61,400 | 130 | 134 | 142 | 107 | 114 | 128 | 3.4 | 2.9 | 2.2 | 95% |
| Asia | Long, wide body | 115 | 8,216 | 270,000 | 259,000 | 245,000 | 425 | 438 | 466 | 350 | 372 | 419 | 3.3 | 2.8 | 2.1 | 97% |

[a]Includes freight, where 100 kg of freight =1 passenger equivalent.

instead assume that the fleet is composed of new aircraft. For example, the 2050 fleet consists only of aircraft manufactured in 2050. This is a simplification that leads to underestimate the fuel consumption of the fleet. Indeed, the age of the European passenger fleet ranges from 9 to 10.5 years old, according to Eurocontrol[75]. At an annual 0.6% improvement in engine efficiency, the fleet, as modeled in this study, is probably underestimating the fuel consumption (and related emissions) by roughly 5%.

### Surface and flight emissions

Surface emissions are based on the life-cycle inventories for the manufacture of aircraft (i.e., including the production of steel and light-weighting materials, such as carbon fiber, aluminum, etc.), airports, the supply of conventional jet fuel and syn-jet fuel (i.e., including the generation of electricity) as well as the operation of the DAC and subsequent $CO_2$ storage. These inventories (and related emissions) are modified over time and across climate scenarios to reflect future policies towards renewables, technological improvements, and learning rates for emerging technologies such as DAC. The source and assumptions behind the fuel supply and DAC operation are detailed in Fuel supply and DAC and $CO_2$ storage. The following surface emissions emitted by the above systems are considered: fossil $CO_2$, methane, hydrogen, HFC-152a, HCFC-140, HCFC-22, HFC-134a, R-10, HFC-125, CFC-11, HFC-143a, and CFC-113.

Flight emissions are calculated following the approach of Cox et al.[29] during take-off, climb, cruise, descent, and landing. They are quantified based on emission factors provided by EMEP/EEA's air pollutant emissions inventory guidebook[74]. Although we model the emissions of 22 different substances, only the following substances are believed to have a potential direct or indirect forcing effect when emitted in the atmosphere[4]: nitrogen oxides, black carbon, sulfur oxides, water vapor, the formation of cirrus clouds as well as fossil $CO_2$. Improvements in emissions for specific substances are from the trends found in the ICAO engine emission database[76]. As such, we consider an annual reduction in the emissions of $NO_x$ and $SO_x$ of 0.6% and 0.3% of black carbon per unit mass of fuel used (which itself also improves by an annual 0.6%). Fuel consumption directly determines water vapor emissions (i.e., ~1.2 kg water vapor per kg jet fuel). Emissions relating to take-off, climb, descent, and landing are all assumed to occur below 9000 m. Cruise emissions are further distinguished between the amount released below and above 9000 m, where 9000 m is used as a threshold value between the end of the troposphere and the beginning of the stratosphere. The amount of fuel consumed below and above 9,000 m is a function of the flight distance and is derived from anonymized air traffic data[77]—see "Share of fuel burnt above 9000 m" under the "Aircraft specs" tab of the spreadsheet model. Such relation defines the share of the emissions during the Cruise phase emitted in the troposphere and stratosphere, respectively. Generally, the longer the flight, the higher the percentage of fuel burnt above 9000 m.

In the upper-troposphere and lower-stratosphere (i.e., up to 9000 m of altitude), "non-$CO_2$" effects triggered by the emissions of $NO_x$ are considered. Above 9000 m, "non-$CO_2$" effects from water vapor emissions, sulfur oxides, and soot are considered. Emissions of $CO_2$ are considered regardless of the altitude. Finally, the warming induced by the formation of persistent contrails and cirrus clouds is calculated based on the distance flown, as indicated in Lee et al.[4].

The hydrogen-to-carbon (H/C) ratio for jet fuel is 1.89, while it is 2.15 for syn-jet fuel[78]. Using the relation in Eq. (1) results in hydrogen content of 13.7% and 15.3% for jet fuel and synthetic jet fuel, respectively.

$$\%H(\text{mass}) = \frac{100\,H/C}{11.916 + H/C} \qquad (1)$$

According to Snijders et al.[78] and Altaher et al.[79], syn-jet fuel leads to a different emission profile than regular jet fuel. We

consider this by applying correction factors on the emission of certain substances—refer to Emission correction factors, under the "Fuel specs" tab of the spreadsheet model. For example, emissions of sulfur oxides and most hydrocarbons, such as volatile organic compounds, are null. In contrast, some others, such as particulate matter (of which soot or black carbon), are significantly reduced. A higher-than-average hydrogen content explains these emission reductions in Fischer-Tropsch fuels with a low aromatics content. The relation between hydrogen content, aromatics content (notably naphthalene), and the reduction of soot particles is described in the work of Voigt et al.[57]. Lower emissions of soot particles reduce the extent of ice nucleation and lead to ice crystals larger in size, which affects the cloudiness and persistence of contrails and cirrus clouds altogether. With a known reduction in ice particle emissions, a corresponding decrease in contrail cirrus clouds is estimated based on ref. [58]. Details on this estimate can be consulted under the tab "Aircraft specs" of the spreadsheet model.

Additionally, the hydrogen content of the jet fuel and syn-jet fuel (i.e., 13.7% and 15.3%, respectively) is used to determine the water vapor emission factor based on the stoichiometric condition that 1 mol of $H_2$ produces 1 mol of $H_2O$ upon combustion. This yields 1.23 kg and 1.39 kg of water vapor per kg of (fossil) jet fuel and syn-jet fuel, respectively.

### Radiative forcing and warming contribution of emissions
Carbon dioxide and SLCF act on different time scales: almost half of the $CO_2$ emitted in the atmosphere accumulates to warm it over several centuries, while the remainder is quickly taken up by oceans (25%[1,80]) and land (29%[1]). SLCF and related effects caused by aviation, on the other hand, act on a smaller time scale with atmospheric lifetimes of hours (e.g., condensation trails and aviation-induced cirrus clouds), days (e.g., sulfate particles and black carbon) or weeks to months (e.g., water vapor)[81]. Therefore, a unified framework considering the different atmospheric lifetimes of species is needed.

The RF of each substance emitted at the surface level is calculated based on several properties. That includes the RF indices provided by the Supplementary Information document of chapter 7 of the IPCC's AR6 WG1 report[54], except for hydrogen, which (indirect) RF index is given by Paulot et al.[60]. These values are reported in Table 3.

The (effective) RF indices of flight emissions and associated uncertainty are sourced from Lee et al.[4], and their atmospheric residence time values are from ref. [81]. Both are also presented in Table 3. Note that the RF of cirrus clouds formation is expressed in reference to the distance flown.

The RF contribution of $CO_2$ and SLCF for each global air-traffic demand trajectory is obtained by multiplying the forcing response matrix of the gas by the mass emitted over time—see Eq. (2)—as per the approach in Allen et al.[7]:

$$f_A = F_A e_A \qquad (2)$$

The forcing response matrix $F$ of the gas $A$ denoted $F_A$ corresponds to a triangular Toeplitz matrix with elements consisting of the first derivative of the absolute global forcing potential (AGFP)—the time-integrated RF of a pulse emission of gas $A$[82]. The determination of the AGFP requires the lifetime and RF index of the gas listed in Table 3. The sum of $F_A$ along columns represents the forcing response of gas $A$ over time because of a pulse emission at year 0, considering its atmospheric residence time. The vector $f_A$ returns the overall RF caused by emissions of gas $A$ for each year of the period considered.

Multiplying $f_A$ by the inverse of the $CO_2$ forcing response matrix gives the amount of $CO_2$ that would cause an equivalent amount of RF—see Eq. (3).

$$e_{CO_2} = F_{CO_2}^{-1} f_A \qquad (3)$$

Allen et al. named $e_{CO_2}$ the Linear Warming-equivalent $CO_2$ emissions ($CO_2$-LWE). This last step is only needed for emissions of SLCF, not $CO_2$.

Multiplying the cumulative sum of emissions of $CO_2$ and $CO_2$-LWE for each SLCF by the transient climate response to cumulative carbon emissions (TCRE) gives the increase in the global mean surface temperature (GMST) over the period considered—see Eq. (4).

$$\Delta T = \chi (CO_2 + CO_2 LWE) \qquad (4)$$

Where $\Delta T$ is the change in global temperature (in Kelvin or degree Celsius) over period $t$ and $\chi$ is the transient climate response to cumulative carbon emissions, which is the ratio of the global average surface temperature change per unit of $CO_2$ emitted and is given a value of 0.45 °C per trillion tons of $CO_2$.

Summing the emissions of $CO_2$ and $CO_2$-LWE for each SLCF gives the total amount of $CO_2$ to capture and sequester by DACCS to mitigate their contribution to global warming.

This approach implies that to stabilize the impact of aviation on the climate with respect to today or to a given time (i.e., to achieve warming neutrality), $CO_2$ emissions should be reduced to net-zero (e.g., through emissions avoidance and offsetting) while emissions of SLCF should be kept constant at the level of the reference time; reducing the latter would have a net-cooling effect with respect to such reference time[6]. On the contrary, to neutralize the impact of aviation on the climate over the whole period (i.e., to achieve *climate-neutrality*), both emissions of $CO_2$ and SLCF should be reduced to net-zero (e.g., through emissions avoidance and offsetting).

### Costs analysis
The cost estimate is based on the methodology and assumptions from Becattini et al.[53]. A simplified system analysis of the different technology options is performed. The various options are compared based on the costs of the jet fuel supply (conventional or synthetic) and of the mitigation technologies (i.e., for DACCS), where applicable. The fuels and DACCS costs considered include the investment and operation costs, which are approximated and reduced to electricity costs. Future costs of low-maturity technologies (i.e., DAC and water electrolysis) are projected using learning curves, with the learning rate depending on the climate and socio-economic scenario considered. Unless otherwise specified, all input values are taken from Becattini et al.[53].

The cost of electricity is an input common to all the technology options considered and is taken from the SSP2 scenarios of the REMIND Integrated Assessment Model v.2.1[83]. In 2030, it is assumed to be 0.064 and 0.074 €/kWh in the 3.5° and 2 °C scenarios, respectively. In 2050, it is considered to decrease to 0.056 and 0.050 €/kWh in the 3.5° and 2 °C scenarios, respectively. Afterward, it is assumed to remain constant.

### Fuel supply
Life-cycle inventories for conventional jet fuel are from the life-cycle inventory database ecoinvent v.3.7.1, "cut off by classification" system model[84]. The entire fuel supply chain includes raw oil extraction and refining to kerosene transport to the airport. The cost of conventional jet fuel is assumed to be 0.50 €/kg, corresponding to the average production cost in 2019[53]. Most recent data are not considered since they were strongly impacted by Covid-19 and the Russia-Ukraine war and did not reflect standard market conditions. We consider a cost of 0.0086 €/kg for conventional and synthetic jet fuel for transporting, distributing, and dispensing the final fuel amount[85].

**Table 3 | Properties for surface and flight emissions**

| Surface emissions | | | | | |
|---|---|---|---|---|---|
| Species | Lifetime [years] | Molecular mass [g/mol] | Radiative eff. [W/m²/ppb] | Radiative eff. [W/m²/kg] | Source |
| Hydrogen | 2.5 | 1.01 | 1.3E-04 | 7.28E-13 | 60 |
| HFC-152a | 1.6 | 66.05 | 0.102 | 8.71E-12 | 54 |
| HCFC-140 | 5 | 133.4 | 0.065 | 2.75E-12 | |
| HFC-32 | 5.4 | 52.02 | 0.111 | 1.20E-11 | |
| HCFC-141b | 9.4 | 116.95 | 0.161 | 7.77E-12 | |
| HCFC-22 | 11.9 | 86.47 | 0.214 | 1.40E-11 | |
| Methane | 11.8 | 16.04 | 3.88E-04 | 1.36E-13 | |
| HFC-134a | 14 | 102.03 | 0.167 | 9.23E-12 | |
| HCFC-142b | 18 | 100.49 | 0.193 | 1.08E-11 | |
| R-10 | 32 | 153.823 | 0.166 | 6.09E-12 | |
| HFC-125 | 30 | 120.02 | 0.234 | 1.10E-11 | |
| CFC-11 | 52 | 137.37 | 0.259 | 1.06E-11 | |
| HFC-143a | 51 | 84.04 | 0.168 | 1.13E-11 | |
| CFC-113 | 93 | 187.375 | 0.301 | 9.06E-12 | |
| Carbon dioxide | | 44.01 | 1.33E-05 | 1.70E-15 | |
| Flight emissions | | | | | |
| Species | Lifetime [years] | Molecular mass [g/mol] | Radiative eff. [W/m²/kg, or W/m²/km for cirrus] | 5th/95th percentile for radiative eff. [W/m²/kg, or W/m²/km for cirrus] | Source |
| NO$_x$ (as NO$_2$) | 11.8 | 46.01 | 1.67E-12 | −2.4E-12/3.83E-12 | 4,81 |
| BC | 0.02 | 12.01 | 5.54E-10 | 7.95E-12/4.29E-10 | |
| SO$_x$ | 0.011 | 64.07 | −1.10E-10 | −4.98E-11/-6.87E-12 | |
| H$_2$O | 0.8 | 18.02 | 2.86E-14 | 2.1E-15/8.3E-15 | |
| Cirrus | 0.00057 | | 9.36E-13 | 6.3E-13/1.39E-12 | |

Life-cycle inventories for the syn-jet fuel supply are based on van der Giesen et al.[86]. They include the synthesis of syngas into different co-products, including jet fuel. The production of syngas relies on the supply of hydrogen and carbon monoxide. The carbon monoxide is supplied via a reverse water-gas shift reaction, using $CO_2$ from the atmosphere (see "Direct air capture and $CO_2$ storage") as input. The hydrogen is supplied by water electrolysis using proton-exchange membrane (PEM) electrolyzers. The allocation of the Fischer-Tropsch synthesis process burden is based on the respective energy content of the co-products. A post-allocation carbon correction of 40 g $CO_2$/kg of syn-jet fuel is considered to ensure that the embodied $CO_2$ per kilogram of syn-jet fuel matches its combustion emission factor of 3.14 kg $CO_2$ per kg fuel. Details about allocating the burden associated with the synthetic gas between the different fuel products can be consulted under the tab "Fuels specs" of the spreadsheet model.

Inventories for the hydrogen production by water electrolysis are from Zhang et al.[87]. We consider a 1% mass loss along the hydrogen supply chain, as per the central estimate found in ref. 88. An improvement factor in terms of energy efficiency is considered, bringing the electricity use per kg of hydrogen from 55 kWh in 2020 down to 44 kWh in 2050[89], besides a fixed amount of electricity (i.e., 3.2 kWh/kg $H_2$) to compress the hydrogen from 25 to 700 bar before storage[90].

The cost of synthetic jet fuel is determined by the cost of DAC, water electrolysis, reverse water-gas shift, Fischer-Tropsch synthesis and upgrading, and fuel distribution. For the costs of DAC, see Direct air capture and $CO_2$ storage. Current levelized investment costs for water electrolysis are assumed to be 0.91 € per kg of hydrogen. In addition to electricity costs (based on the electricity use mentioned above stemming from the electrolyzer operations and hydrogen compression), we account for the cost of water purification, which is, however, marginal (ca. 0.0084 €/kg$H_2$[91]). Projected investment costs for water electrolysis are estimated for 2030 and 2050, assuming a

learning rate of 5% and 20% for the 3.5 °C and 2 °C climate scenario, respectively, and a time-dependent cumulative capacity as in Becattini et al.[53]. The resulting levelized costs (LC) for $H_2$ production and compression estimated for 2030 and 2050 are summarized in Table 4.

The total levelized investment cost for the reverse water-gas shift, the Fischer-Tropsch synthesis, and further upgrading is assumed to be 0.56 €/kg of synthetic jet fuel[85]. The electricity costs for these steps are also accounted for using the electricity prices indicated in *Costs analysis*. Table 4 summarizes the levelized cost of synthetic jet fuel supply estimated for 2030 and 2050.

**Direct air capture and CO$_2$ storage**

Life-cycle inventories for the supply of $CO_2$ via direct air capture (DAC) are from Terlouw et al.[92] using the configuration connected to the electricity grid. A heat pump assists the operation of DAC with a coefficient of performance of 2.9. Expected improvements and learning rates regarding energy use of the DAC system are from Hanna et al.[93]. A learning rate of 2% is considered for heat and electricity use, applied to the deployment capacities projected in ref. 94, where 5 Gt $CO_2$ are annually captured by 2050. It is to note that there is a risk of inconsistency as the deployed capacities do not change with the climate scenario considered. In other words, the operational efficiency assumed for DAC might be overestimated in the 3.5 °C scenario. There is also a fixed amount of electricity (i.e., 0.2 kWh/kg $CO_2$ captured) considered to compress the gas from 1 to 100 bar before injection in the reverse water-gas shift reactor (if reused) or the pipeline for subsequent geological storage. For cases where the $CO_2$ is subsequently reused to produce syn-jet fuel, we consider a rather tightly integrated system where the DAC unit is located less than a kilometer away from the fuel producer, thereby minimizing the transport (and loss) of $CO_2$.

In cases where the $CO_2$ is instead stored underground, it is first transported over 400 km by pipeline and injected 3000 m underground. The inventories for $CO_2$ storage are from Volkart et al.[95]. They

**Table 4 | Levelized costs (LC) for H₂ and synthetic fuel production and compression estimated for 2030 and 2050 for the 2 ° and 3.5 °C climate scenarios**

| Climate scenario | LC H$_2$ (€/kgH$_2$) 2030 | LC H$_2$ (€/kgH$_2$) 2050 |
|---|---|---|
| 2 °C | 4.3 | 2.7 |
| 3.5 °C | 4.7 | 3.4 |
| | LC syn-jet fuel (€/kgJF) 2030 | LC syn-jet fuel (€/kgJF) 2050 |
| 2 °C | 3.1 | 2.1 |
| 3.5 °C | 3.8 | 3.2 |

**Table 5 | Levelized costs for direct air capture (DAC) and direct air capture and carbon storage (DACCS) estimated for 2030 and 2050 for the 2° and 3.5 °C climate scenarios**

| Climate scenario | LC DAC (€/kgCO$_2$) 2030 | LC DAC (€/kgCO$_2$) 2050 |
|---|---|---|
| 2 °C | 0.15 | 0.10 |
| 3.5 °C | 0.40 | 0.37 |
| | LC DACCS (€/kgCO$_2$) 2030 | LC DACCS (€/kgCO$_2$) 2050 |
| 2 °C | 0.16 | 0.12 |
| 3.5 °C | 0.41 | 0.38 |

include the site preparation and borehole drilling, the pipeline transport of the $CO_2$ to the injection site, minor leakage during transport, and the necessary compressors and electricity for $CO_2$ compression.

The current levelized investment cost of DAC is estimated to be 0.54 €/kgCO$_2$[94]. The electricity expenditures associated with DAC operations and $CO_2$ compression are computed based on the above-mentioned electricity prices. Projected investment costs for DAC are estimated for 2030 and 2050, assuming a learning rate of 5% and 20% for the 3.5° and 2° climate scenario, respectively, and a time-dependent cumulative capacity as in Becattini et al.[53]. The resulting levelized costs for $CO_2$ from DAC (including $CO_2$ compression) estimated for 2030 and 2050 are summarized in Table 5.

Where applicable, a levelized cost of 0.02 €/kg $CO_2$ is assumed to transport $CO_2$ by pipeline over 400 km. The levelized investment cost for $CO_2$ underground storage is 0.01 €/kg $CO_2$, while electricity expenditures associated with $CO_2$ injection are computed based on a consumption of 0.0334 kWh/kg $CO_2$.

Table 5 summarizes the levelized cost of DACCS estimated for 2030 and 2050 under the 2° and 3.5 °C climate scenarios.

**Prospective life-cycle inventory database**
The life-cycle inventory database ecoinvent v.3.7.1[96] is used as a "background" database to quantify the emissions of $CO_2$ and SLCF related to the provision of commodities and services necessary to the different life-cycle phases of the flight (e.g., electricity, heat, metals, road) that are not explicitly modeled in the spreadsheet. While this database is apt to represent commodities' current supply chain performance, it is transformed to perform prospective analyses (i.e., future flight performances in 2030 and 2050). To do so, the Python library premise[96] is used with scenario projections of the Integrated Assessment Model REMIND v.2.1[83,97]. premise adjusts several aspects of the ecoinvent database to align with the scenario projections of REMIND, across different Representative Concentration Pathways (RCP), given the Shared Socio-economic Pathway SSP 2 (also called Middle of the road). The SSP 2 can roughly be described as an extrapolation into the future of the historical economic and societal development observed until now. In the cases presented, the scenarios equivalent to SSP 2 with RCP 2.6 and RCP 6 represent the 2 °C and 3.5 °C climate scenarios, respectively. We refer the reader to ref. 98 for a detailed explanation of how SSPs relate to RCPs. On this basis, the electricity, steel, cement, and transport sectors of the ecoinvent v.3.7 database have been modified to reflect investments and efficiencies described in the REMIND scenarios. As shown in ref. 96, the electricity sector is most influential in the context of prospective LCA in general and even more so in our analysis, as the contributions from European electricity supply dominate the release of GHG associated with syn-jet fuel production and DACCS. We provide scenario-specific market shares and related $CO_2$ emissions of the European electricity consumption mix over time in Table 6. Note that Table 6 shows uncharacterized emissions of $CO_2$ only, not the characterized contribution of all GHG (although fully accounted for in the model). The share of electricity from fossil fuels is below 1% in the 2 °C scenario in 2030 and

thus in line with the latest EU directive on GHG emission savings of renewable liquid and gaseous transport fuels of non-biological origin and from recycled carbon fuels[51]. An essential difference between our rigorous LCA approach and the EU directive is the fact that while we do quantify emissions associated with the production of capital goods (e.g., wind turbines and PV panels), the EU directive does not and instead it attributes zero GHG emissions to electricity from wind, solar, geothermal and hydropower. Further, the directive does not include GHG emissions emitted from reservoirs of hydropower plants, which can be substantial[99,100]—we do include those emissions. Consequently, climate impacts of power generation remain positive, even if they are reduced substantially. From a rigorous LCA perspective, a truly net-zero economy would have to eliminate the use of fossil fuels entirely and compensate for remaining GHG emissions by removing equivalent amounts of $CO_2$ from the atmosphere. We have not taken such a scenario into account but consider it as an important subject for further research.

The technology development for electrolysis, Fischer-Tropsch synthesis, and direct air capture of $CO_2$, together with the dynamic modeling of the life-cycle background database (as detailed in the sections above), results in a substantial decrease in emissions of GHG associated with the production of syn-jet fuel production over time. Table 6 provides the resulting life-cycle-based $CO_2$ emissions per kg of syn-jet fuel—combustion-related $CO_2$ emissions of 3.14 kg $CO_2$/kg syn-jet fuel, equal to the amount of $CO_2$ extracted from the atmosphere to produce the CO needed for the Fischer Tropsch process, have been added. Note that Table 6 shows uncharacterized emissions of $CO_2$ only, not the characterized contribution of all GHG (although fully accounted for in the model).

For reference, the life-cycle $CO_2$ emissions of (fossil) jet fuel production and combustion are ~3.57 kg $CO_2$/kg, with little reduction over time. The figures provided here must not be directly compared with the recent EU directive on GHG emission savings of renewable liquid and gaseous transport fuels of non-biological origin and from recycled carbon fuels[51], since the GHG emission reduction threshold of 70% required according to the directive is based on zero GHG emissions from non-biomass renewable power, while we quantify complete life-cycle emissions.

**Resource impact modeling for European fleet scenarios**
The prospective life-cycle inventory database represents the basis for our quantification of freshwater abstraction and land use shown in Fig. 4, with synthetic jet fuel assumed to be produced at generic European locations. Freshwater abstraction requirements represent the uncharacterized (i.e., not considering site-specific scarcity) cumulative flows of freshwater withdrawal (i.e., not considering the fate of the freshwater) in terms of volume in each of our scenarios. Land use represents the uncharacterized (i.e., not considering site-specific land quality) cumulative occupied area of urban and agricultural land over time (i.e., m² over a year) in each of our scenarios.

The cumulative use of electricity, land, and freshwater of the European fleet over 2018–2100 is compared to the current consumption levels in the European Union. The amount of electricity currently

**Table 6 | Life-cycle CO$_2$ emissions of the European electricity consumption mix and syn-jet fuel production in the selected climate scenarios over time**

| | RCP 6 (3.5 °C scenario) | | | | | RCP 2.6 (2 °C scenario) | | | | |
|---|---|---|---|---|---|---|---|---|---|---|
| | Power generation | | | | | | | | | |
| Share in consumption mix | 2020 | 2030 | 2040 | 2050 | 2100 | 2020 | 2030 | 2040 | 2050 | 2100 |
| Biomass CHP | 2.4% | 2.0% | 1.4% | 0.5% | 0.0% | 2.4% | 3.0% | 2.0% | 1.2% | 0.0% |
| Biomass IGCC CCS | 0.0% | 0.0% | 0.0% | 0.0% | 0.0% | 0.0% | 0.3% | 0.3% | 0.3% | 0.0% |
| Biomass IGCC | 1.7% | 1.3% | 0.7% | 0.1% | 0.0% | 1.7% | 1.5% | 0.9% | 0.3% | 0.0% |
| Coal PC | 14.3% | 6.0% | 2.8% | 0.4% | 0.0% | 14.3% | 0.0% | 0.0% | 0.0% | 0.0% |
| Coal IGCC | 0.0% | 0.0% | 0.0% | 0.0% | 0.0% | 0.0% | 0.0% | 0.0% | 0.0% | 0.0% |
| Coal PC CCS | 0.0% | 0.0% | 0.0% | 0.0% | 0.0% | 0.0% | 0.0% | 0.0% | 0.0% | 0.0% |
| Coal IGCC CCS | 0.0% | 0.0% | 0.0% | 0.0% | 0.0% | 0.0% | 0.0% | 0.0% | 0.0% | 0.0% |
| Coal CHP | 5.0% | 2.8% | 1.3% | 0.2% | 0.0% | 5.0% | 0.0% | 0.0% | 0.0% | 0.0% |
| Gas OC | 0.6% | 0.3% | 0.1% | 0.0% | 0.0% | 0.6% | 0.3% | 0.0% | 0.0% | 0.0% |
| Gas CC | 15.6% | 17.4% | 11.9% | 6.6% | 0.0% | 15.6% | 0.0% | 0.0% | 0.0% | 0.0% |
| Gas CHP | 0.0% | 0.0% | 0.0% | 0.0% | 0.0% | 0.0% | 0.0% | 0.0% | 0.0% | 0.0% |
| Gas CC CCS | 0.0% | 0.0% | 0.0% | 0.0% | 0.0% | 0.0% | 0.0% | 0.0% | 0.0% | 0.0% |
| Geothermal | 0.3% | 0.9% | 0.9% | 0.8% | 0.6% | 0.3% | 0.9% | 0.7% | 0.7% | 0.5% |
| Hydro | 12.5% | 11.7% | 10.4% | 9.6% | 7.4% | 12.5% | 11.1% | 8.9% | 8.2% | 5.6% |
| Nuclear | 29.8% | 20.6% | 12.1% | 5.3% | 0.0% | 29.8% | 19.1% | 9.8% | 4.3% | 0.0% |
| Oil ST | 1.5% | 0.3% | 0.0% | 0.0% | 0.0% | 1.5% | 0.3% | 0.0% | 0.0% | 0.0% |
| Solar CSP | 0.2% | 0.2% | 0.1% | 0.0% | 0.0% | 0.2% | 0.1% | 0.1% | 0.0% | 0.0% |
| Solar PV Centralized | 4.9% | 14.4% | 23.8% | 26.1% | 27.1% | 4.9% | 26.5% | 30.9% | 29.8% | 29.2% |
| Wind Onshore | 9.6% | 18.0% | 28.0% | 36.9% | 37.9% | 9.6% | 31.2% | 37.4% | 38.3% | 37.1% |
| Wind Offshore | 1.6% | 4.2% | 6.5% | 13.5% | 27.0% | 1.6% | 5.8% | 8.9% | 16.9% | 27.6% |
| CO$_2$ intensity of electricity consumption mix [g CO$_2$/kWh, incl. grid losses] | 383 | 302 | 218 | 130 | 27 | 383 | 89 | 32 | 27 | 20 |
| Total amount of electricity generation [EJ] | 12 | 13 | 14 | 14 | 17 | 12 | 13 | 16 | 22 | 32 |
| | Syn-jet fuel production | | | | | | | | | |
| CO$_2$ intensity of syn-jet fuel production [kg CO$_2$/kg, incl. distr. losses] | 3.9 | 3.19 | 2.66 | 2.14 | 1.71 | 3.9 | 1.89 | 1.52 | 1.48 | 1.45 |

*CHP* combined heat power plant, *IGCC* integrated gasification combined cycle power plant, *CCS* carbon capture and storage, *PC* pulverized coal, *OC* organic (rankine) cycle, *CC* combined cycle, *ST* steam turbine, *CSP* concentrated solar power, *PV* photovoltaic.

produced and consumed in the European Union in 2020 (i.e., EU-28) was approximately 2600 TWh, as reported by eurostat[62]. The total surface area of the European Union is about 4.2 million km$^2$, according to ref. 101. The area in the EU-28 used for farming purposes was 173 million hectares in 2016, according to Eurostat[102]. The annual amount of freshwater abstracted in the EU-28 annually, all sectors considered, is around 350 km$^3$, according to ref. 103. It is estimated that an average European citizen uses about 144 liters of freshwater per day (or about 52 m$^3$ per year)[104]. Finally, the Gross Domestic Product of the European Union in 2019 was 17 trillion USD[105], considering 1 € = 1 USD.

**Reporting summary**
Further information on research design is available in the Nature Portfolio Reporting Summary linked to this article.

## Data availability
The results graphed in this study are available from https://doi.org/10.5281/zenodo.8059750.

## Code availability
The data input, model, and scripts used in this study are available from https://doi.org/10.5281/zenodo.8059750.

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

## Acknowledgements
This work has received financial support from the SynFuel initiative of the ETH Council, PSI's Energy System Integration platform, and the Kopernikus-Projekt Ariadne (FKZ 03SFK5A), funded by the German Federal Ministry of Education and Research. Further financial support has been provided via the research project SHELTERED, funded by the Swiss Federal Office of Energy (SFOE).

## Author contributions
Conceptualization: R.S., V.B., C.B., M.M.; Formal analysis: R.S., V.B., P.G.; Funding acquisition: C.B., M.M.; Investigation: R.S., V.B., P.G., B.C.; Methodology: R.S., V.B., P.G., A.D.; Software: R.S., V.B., A.D.; Supervision: C.B., M.M.; Visualization: R.S., V.B., M.M.; Writing—original draft: R.S., V.B.; Writing—review and editing: R.S., V.B., P.G., C.B., M.M., A.D., and B.C.

## Competing interests
The authors declare no competing interests.
