## [Peer review file · Nature Communications]

REVIEWER COMMENTS

Reviewer #1 (Remarks to the Author):

The paper «climate-neutral aviation: will it fly?» addresses a highly topical question using a comprehensive LCA approach. Whereas several other studies use LCA to explore the impacts of different aviation technologies and aviation fuels, this study is noteworthy for focusing on the overall impacts of the sector in climate neutrality scenarios. It is particularly important that the study focuses on the totality of the impacts from the sector as a whole, in particular the total pressure on economic and natural resources, rather than taking a marginal or relative approach as most studies do. It is also laudable that the study devotes sufficient attention to the (too often ignored) non-CO₂ impacts. I furthermore appreciate that the different properties of the climate forcers are accounted for using the Linear-Warming-Equivalent method. Overall, this is a very thorough study, reflected in details such as including H₂ leakage and the impact this has via extending the atmospheric lifetime of methane. I only have one major concern with this study, and that is the lack of a sensitivity or robustness analysis. A complex study such as this relies on a very large number of specific assumptions, and it is very difficult to assess how robust the findings are without some type of systematic sensitivity analysis with respect to key (uncertain) factors, including the non-CO₂ climate impact, future improvements in efficiency, etc.

The only other major comment I have is that the main conclusion is a bit too broad given the analysis. The study concludes that “the idea of a climate-neutral aviation will fly only if air traffic decreases to reduce the scale of climate impacts to mitigate.” I would agree that the study shows that if the aviation industry relies on SAF and DAC alone, it is not possible to become climate neutral without exerting excessive pressure on resources (but the conclusion appears to be broader than this). The two mitigation options considered are not the only mitigation options, and they can be combined with others. One potentially important omission is navigational avoidance: This is a highly uncertain and unproven mitigation option today, but it can potentially – through minimal increases in energy use – avoid a very large proportion of the contrail formation, resulting in a much reduced demand for DAC to offset the remaining climate impact. (The same would hold if the climate impact from contrail cirrus turns out to be at the lower end of current estimates, something a sensitivity analysis would likely reveal). I am not suggesting that the authors necessarily include further mitigation options in their analysis, but that the existence of other mitigation options, and uncertainties, are better reflected and accounted for in the conclusions.

Minor issues:

The abstract claims that reducing CO₂ emissions alone would leave up to 80% of the climate impacts unaccounted for. What is the basis for this? I cannot find that number anywhere else in the paper. One assumption in particular surprised me – a carbon intensity of 30 gCO₂/kWh in 2100. I found it a little difficult to track down the origin of this assumption (I might have overlooked something), but it seems inconsistent with the projected rapidly declining emissions intensity of electricity projected until 2050 (trend lines would seem to hit the zero line relatively soon thereafter). Again, this is something that can be accounted for through a sensitivity analysis. The authors claim that “Although associated with large uncertainties, the understanding of non-CO₂ effects has improved over the years, allowing characterization of the relationship between atmospheric SLCF emissions and increase in radiative forcing (RF) with acceptable accuracy...” I’m not sure I would agree. The Lee et al. paper cited in defense of this claim concludes that “Uncertainty distributions (5%, 95%) show that non-CO₂ forcing terms contribute about 8 times more than CO₂ to the uncertainty in the aviation net ERF in 2018. The best estimates of the ERFs from aviation aerosol-cloud interactions for soot and sulfate remain undetermined.” I’m also not sure the uncertainty has decreased over time. (The authors also discuss these uncertainties in some detail on page 7).

Reviewer #2 (Remarks to the Author):

Right now, for every \$100 that an airline spends to operate a flight, about \$20 goes to purchase the needed fuel, and about \$6 goes for the cost of purchasing and maintaining the airplane. A new medium-sized airplane costs about \$30 million. So that means that for every one of these airplanes in operation, something like \$100 million is being spent, somewhere, to build and operate the energy infrastructure needed to supply it with fuel. Right now that is a gigantic global network of oil wells, pipelines, ships, and refineries. If you were to instead spend those \$100 million on solar PV panels, the retail cost of which is about \$50 per square meter and which require virtually no money to operate, it means that you could cover 200 hectares of land. Wow! That's a lot, occupying far more space than the plane itself. But it's the way the world works.

Meanwhile, other stuff is happening. First, mainly because the cost of flying relative to people's incomes has been falling, the amount people fly – and the investments into new planes and new energy infrastructure – has been growing, at something like 2.5% per year. If this trend continues, which airlines are planning on, the number of planes in the sky could triple by 2050. Second, most countries have decided to achieve net-zero emissions, effectively phasing out the use of fossil fuels across their economies. In Europe, the plan for the aviation sector, consistent with the net-zero economy-wide target, is to make sure that by 2065 all of the planes in operation are burning fuel derived from renewable sources, like solar power. To achieve the net-zero climate target they may also have to start requiring offsetting the non-CO2 emissions from flying. We don't know if these two planned trends – the growth in aviation volume and the move towards net-zero emissions – are compatible with each other. It probably depends on how much it costs to produce the renewable fuel and pay for the offsetting. If it costs about the same as it does to produce fossil kerosene, then the growth trend may continue. If it were to cost three times as much, then instead of \$20 going for fuel, airlines would have to spend \$60 for fuel and offsetting, pushing up their total costs by 40%. Given observed price elasticities, that would reduce demand for flying by about the same amount – 40% – meaning that instead of air traffic volume growing to 300% of its current level, it would grow to only 180%. If the cost of the fuel and offsetting were even higher, which would likely be the case if there were severe resource constraints, then the growth in aviation volume would be even less, perhaps even becoming negative.

The authors of this paper examine whether there would be severe resource constraints, going through a great deal of analysis to reach the conclusion that there would be, and that the decarbonization plans are inconsistent with the aviation growth plans. The problem is that where they actually compare what would be needed with what is available, they aren't doing so for resources. Mainly they compare the electricity that would be needed for the fuels production to serve Europe with the electricity Europe currently produces. Electricity is not a fixed resource. It currently represents a rather small share of the energy system, since we use electricity for fairly small stuff, things like lighting, refrigeration, and computing. As we start using electricity for the big stuff, like driving cars and flying planes and heating our buildings, it is clear that investment flows will have to shift from fossil infrastructure to electricity production infrastructure, like those 200 hectares of solar panels that could be bought for each plane in the sky if we were to redirect investment.

Of course there are resource constraints that could actually constrain this needed growth in electricity production: water, land, materials, and labour. How these constraints play out depends, of course, depends on the technology used for generating electricity. If the promises people make of new nuclear reactors or cold fusion become reality by the second half of the century, then the resource needs will be quite small. Personally I am sceptical of this, and think it is more likely that we will get the electricity from solar PV, CSP, and wind power.

Previous work has shown that there clearly is enough land, assuming one uses land that is most suited, such as in the various deserts of the world. The authors of this paper suggest we would need about 4,160 TWh of electricity production per year. If you put that solar power in a place with a GHI of over 2,000 kWh / m², and assume a solar conversion efficiency of 20%, that means that you would

need to cover a little over 10'000 square kilometres of land. That is less than 0.5% of the area of Saudi Arabia, a country that is going to have to figure out how to replace the likely collapse in oil export revenues.

Previous work has also shown that water is not a constraint. Thinking purely about energy production, Damerau et al. (Climatic Change, 2015) compared the water requirements for operating CSP plants (which relative to PV needs a lot of water), and found these to be less than half of the water needed to extract and refine oil with the same energy content. If we think about the water needing to be electrolyzed for the fuels production, we know that the current state-of-the air DAC technology extracts more than this from the air – even arid air in desert regions – simply as an unintended by-product of CO2 capture.

Materials could be a constraint, although the authors do not examine this. Similarly, they do not examine whether labour availability could be a constraint. In either case, though, we need to think about the electricity needs of aviation in the context of the electricity needs of the entire economy. The decarbonization of road transport, for example, will require the switch to electric vehicles (which, one might be tempted to say, is clearly impossible, since the required volume of battery production for cars alone would exceed all the lithium ion batteries we are currently producing, for things like mobile phones and laptop computers), and this in turn will require vastly more electricity than what will be needed for the much smaller aviation sector. Really, the question is whether decarbonization of the economy will fly, where aviation represents one small piece of this. It is the overall decarbonization that will determine whether we push up against resource constraints. And those resource constraints will, partly, influence costs, which in turn will have a feedback on demand. Meanwhile, my understanding of the literature is that, currently, we do not view either labour supply or material reserves to be a major constraint. For example, the Transport chapter in the most recent IPCC report investigated the issue of material constraints for the electrification of ground transport – where many argue the issue is most acute – and did not find them to be a problem.

The authors of this paper convincingly show that what is needed to decarbonize aviation is breathtakingly large, if aviation demand continues to grow at the same rate it has grown before there were any efforts to decarbonize it, and costs were consistently declining. That is important to recognize. The modelling framework they have developed is impressive, and could be useful for estimating the effects of the planned decarbonization efforts on needed investment flows and materials usage. At the same time, one needs to recognize that there are huge error bars. For example, they assume the price of electricity will decline to about €50 per MWh by 2050, and then stay there. Others, such as Way et al. (Joule, 2022) suggest that even by 2030 the cost could well be below €10 per MWh.

While it pains me, I am going to recommend rejection of this paper. The work that has gone into it is impressive. But the way that they think about answering their question – whether we can decarbonize aviation – is fundamentally misleading.

Reviewer #3 (Remarks to the Author):

The work proposed by the authors is significant and interesting. It addresses an interesting topic which is prospective scenarios for climate-neutral aviation. The approach taken is relatively clear and the supplementary materials are significant and useful. Overall I think it is an interesting contribution for the community.

However, I have a few remarks given below but my main ones concern:

- the scope of the paper has to be more underlined especially in the abstract (maybe in the title?): European aviation
- the choice of considering only efuels makes it difficult to compare to other scenarios of the literature.

Hence, I would underline that the conclusions of the paper are given with strong assumptions from the authors.

- the annual improvement of fleet efficiency (+0.6%) seems low to me, I would update that value with values of the proposed references. However, this will not change the conclusions regarding the need for traffic decrease.

Therefore, I would suggest to accept this paper after minor revisions.

Abstract

The scope of the paper should be underlined: European aviation.

1 – Introduction

It would be interesting that at this point of the paper to underline the differences between biofuels and efuels as they imply different primary energy (biomass and low-carbon electricity) and do not have the same technological maturity. For biofuels and efuels I could suggest these references:

Zhao, X., Taheripour, F., Malina, R., Staples, M. D., & Tyner, W. E. (2021). Estimating induced land use change emissions for sustainable aviation biofuel pathways. *Science of the Total Environment*, 779, 146238.

Staples, M. D., Malina, R., Suresh, P., Hileman, J. I., & Barrett, S. R. (2018). Aviation CO2 emissions reductions from the use of alternative jet fuels. *Energy Policy*, 114, 342-354.

Even if the paper focuses on efuels, most of the scenarios in the academic or institutional literature show the usage of biofuels before efuels.

Also regarding recent literature of prospective scenarios that include the climate impact of non-CO2 effects I think that these three articles would have their place in this introduction:

Planès, T., Delbecq, S., Pommier-Budinger, V., & Bénard, E. (2021). Simulation and evaluation of sustainable climate trajectories for aviation. *Journal of Environmental Management*, 295, 113079.

Grewe, V., Gangoli Rao, A., Grönstedt, T., Xisto, C., Linke, F., Melkert, J., ... & Dahlmann, K. (2021). Evaluating the climate impact of aviation emission scenarios towards the Paris agreement including COVID-19 effects. *Nature Communications*, 12(1), 1-10.

Klöwer, M., Allen, M. R., Lee, D. S., Proud, S. R., Gallagher, L., & Skowron, A. (2021). Quantifying aviation's contribution to global warming. *Environmental Research Letters*, 16(10), 104027.

Nevertheless, there are many relevant references and I think it is really relevant to mention the report on Aviation's "missed" climate targets.

2 – Main

In this section the authors present three scenarios that they study. The choices they make seem relevant and for some original.

Regarding climate objectives it would be worth to underline that the Paris Agreement aims to remain well below 2°C.

2.1 - Conventional jet fuel vs. Synthetic jet fuel

It is worth mentioning that including the RF caused by DACCS is really original.

From what I understand synthetic jet fuels only include efuels. It would be worth to mention that the scenarios presented consider that this technology is not yet mature and even 1% in 2026 as considered seems for me very ambitious.

2.2 - Climate-neutral European aviation and resources

It is assumed that the production of efuels are in Europe. Why not in a second part assume it is produced somewhere more appropriate? The least would be to underline that it is unfeasible at European scale.

3 – Discussion

Discussions are interesting but maybe lack of concrete quantified indicators regarding the amount of CDR or energy. This would help justify words like "major" "intensive". Furthermore, I think it is

important to mention that these conclusions rely on assumptions of the studied scenarios and pros of the aviation sector may underline that improvement in efficiencies are low (%0.6 per year) and that biofuels are not considered.

4 – Methods

Generally, this section misses quantified examples and indicators. I encourage the authors to use more of the supplementary data in the text.

4.1 - General workflow

4.2 - Life-cycle environmental and cost modelling

The authors use a life-cycle inventory database that is prospective which makes the work relatively original.

4.3 - European fleet scenarios and demand projection

4.4 – Aircrafts

0.6 % is low compared to 2%-2.5% historic mentioned in the introduction. The Destination 2050 values are quite delicate to manipulate because they also introduce in 2035 "Improved technology (hydrogen)" that contains energy efficiency also. In the approach chosen by the authors there is no coefficient for propulsive efficiency that is more significant than aerodynamics and weight. I think that improvement in efficiency should be refined to probably reach >1% as in Planès et al, Grewe et al, and Klower et al articles. Nevertheless, this should not change conclusions but provide more credibility to the work and results.

In addition, improvement in operations and load factors could be mentioned and discussed.

4.5 - Surface and flight emissions

4.6 - Radiative forcing and warming contribution of emissions

Table 3 and 4 are interesting data and the approach chosen for non-CO2 effects based on Allen et al is clear. However, it would be interesting to compare with approach from Planès et al, Grewe et al, and Klower et al as there are no consensus yet on the approach.

4.7 - Costs analysis

4.8 - Fuel supply

Something compels me in this section. A single value is given for the content of CO2 in a kg of syn-fuel. This depends strongly on the emission factor of electricity which (hopefully) should decrease with time. What electricity grid is considered? What about dedicated grid? Why is it not prospective as for the life-cycle inventory database?

5 - Direct air capture and CO2 storage

6 - Prospective life-cycle inventory database

I think I have my answer for 4.8 but this means I is not straightforward to understand and should be explained more thoroughly in 4.8.

Response to Reviewers

Dear reviewers,

We thank you for the constructive feedback and valuable comments – addressing them and revising the manuscript accordingly certainly improved the quality of the manuscript. Key elements of our revision – following the advice of the editor – are the following:

- We performed and added a comprehensive sensitivity analysis to evaluate uncertainties and potential variability of results to test their robustness (provided in a separate Supporting Information document).
- We revised our conclusions and recommendations to match the scope of our analysis and avoid “overselling”.
- We revisited our limitations section and added further relevant issues.
- We included the latest literature on the topic and now provide a concise comparative overview to stress the comprehensiveness and novelty of our work.

Below we list our point-by-point replies and actions. Besides we have made a careful revision of the text to eliminate any typos. The changes to the manuscript are marked in red in our response.

Reviewer #1:

The paper «climate-neutral aviation: will it fly?» addresses a highly topical question using a comprehensive LCA approach. Whereas several other studies use LCA to explore the impacts of different aviation technologies and aviation fuels, this study is noteworthy for focusing on the overall impacts of the sector in climate neutrality scenarios. It is particularly important that the study focuses on the totality of the impacts from the sector as a whole, in particular the total pressure on economic and natural resources, rather than taking a marginal or relative approach as most studies do. It is also laudable that the study devotes sufficient attention to the (too often ignored) non-CO₂ impacts. I furthermore appreciate that the different properties of the climate forcers are accounted for using the Linear-Warming-Equivalent method. Overall, this is a very thorough study, reflected in details such as including H₂ leakage and the impact this has via extending the atmospheric lifetime of methane.

Comment 1: *I only have one major concern with this study, and that is the lack of a sensitivity or robustness analysis. A complex study such as this relies on a very large number of specific assumptions, and it is very difficult to assess how robust the findings are without some type of systematic sensitivity analysis with respect to key (uncertain) factors, including the non-CO₂ climate impact, future improvements in efficiency, etc.*

Reply: We agree with the reviewer on the importance and the added value of a sensitivity analysis. Following the reviewer’s comment, we have performed

sensitivity analyses on the main parameters influencing our results. The outcome of the analyses is presented in the Supplementary Information; implications are discussed in the *Climate-neutral European aviation and resources* section.

Comment 2: *The only other major comment I have is that the main conclusion is a bit too broad given the analysis. The study concludes that “the idea of a climate-neutral aviation will fly only if air traffic decreases to reduce the scale of climate impacts to mitigate.” I would agree that the study shows that if the aviation industry relies on SAF and DAC alone, it is not possible to become climate neutral without exerting excessive pressure on resources (but the conclusion appears to be broader than this). The two mitigation options considered are not the only mitigation options, and they can be combined with others. One potentially important omission is navigational avoidance: This is a highly uncertain and unproven mitigation option today, but it can potentially – through minimal increases in energy use – avoid a very large proportion of the contrail formation, resulting in a much reduced demand for DAC to offset the remaining climate impact. (The same would hold if the climate impact from contrail cirrus turns out to be at the lower end of current estimates, something a sensitivity analysis would likely reveal). I am not suggesting that the authors necessarily include further mitigation options in their analysis, but that the existence of other mitigation options, and uncertainties, are better reflected and accounted for in the conclusions.*

Reply: We agree with the reviewer on the points raised regarding the scope and breadth of the analysis. We have revised the manuscript accordingly, thus stressing in the *Discussion* section that our conclusions are relative to the mitigation options considered in our work. Furthermore, we have better formulated in the *Main* section the paragraph mentioning other mitigation options that could be adopted by the aviation sector but are not considered in this study, among which navigational avoidance. Finally, in the newly added sensitivity analysis in the Supplementary Information, we show the effect of decreasing the radiative efficiency index (i.e., climate impact) associated with forming cirrus clouds to 14% of the value used in our reference analysis, as hypothesized in a recent study. Such a substantial reduction indeed has a major impact on the results of the analysis, especially for the fossil kerosene case: costs and resource requirements are reduced by up to almost 50%.

Minor issues:

Comment 3: *The abstract claims that reducing CO₂ emissions alone would leave up to 80% of the climate impacts unaccounted for. What is the basis for this? I cannot find that number anywhere else in the paper.*

Reply: As indicated by the reviewer, this quantitative information was not clearly reported in the article, other than in the abstract. We have modified the section *Conventional jet fuel vs. Synthetic jet fuel* to explain how this percentage was calculated:

“Mitigating solely flight-CO₂ emissions would require a significantly smaller CDR effort but **would result in a decrease of only 20% of the GMST with respect**

to the unmitigated scenario (+ 0.028 °C vs. + 0.035 °C, Figure 2.d), thus leaving up to 80% of the climate change impacts unmitigated.”

Comment 4: *One assumption in particular surprised me – a carbon intensity of 30 gCO₂/kWh in 2100. I found it a little difficult to track down the origin of this assumption (I might have overlooked something), but it seems inconsistent with the projected rapidly declining emissions intensity of electricity projected until 2050 (trend lines would seem to hit the zero line relatively soon thereafter). Again, this is something that can be accounted for through a sensitivity analysis.*

Reply: In the 3.5°C climate scenario, reported in the main article, the integrated assessment model projection for electricity generation corresponds to life-cycle emissions of 27 gCO₂/kWh in 2100. It is important to note that approaching zero in the electricity sector from a life cycle perspective would only be possible, if emissions embodied in materials would be entirely eliminated (corresponding to a worldwide complete phase out of fossil fuel use), or if remaining emissions would be compensated for by negative CO₂ emissions generating electricity from biomass with CCS.

Under the 2 °C climate scenario, reported in the Extended Data Figures, a slightly cleaner electricity mix (i.e., 20 gCO₂/kWh in 2100) is projected, consistent with the assumption of a stronger commitment to meet climate goals by the entire society. For both scenarios, the CO₂ intensity of the electricity is a result of integrating scenario projections of the Integrated Assessment Model REMIND into the life-cycle database ecoinvent, as explained in the *Methods* section.

Furthermore, in the newly added sensitivity analyses, we have assessed the effect of using nuclear electricity (with a CO₂ intensity of 6 gCO₂/kWh) to power the different technologies considered (mainly synthetic jet fuel production and DACCS).

Comment 5: *The authors claim that “Although associated with large uncertainties, the understanding of non-CO₂ effects has improved over the years, allowing characterization of the relationship between atmospheric SLCF emissions and increase in radiative forcing (RF) with acceptable accuracy...” I’m not sure I would agree. The Lee et al. paper cited in defense of this claim concludes that “Uncertainty distributions (5%, 95%) show that non-CO₂ forcing terms contribute about 8 times more than CO₂ to the uncertainty in the aviation net ERF in 2018. The best estimates of the ERFs from aviation aerosol-cloud interactions for soot and sulfate remain undetermined.” I’m also not sure the uncertainty has decreased over time. (The authors also discuss these uncertainties in some detail on page 7).*

Reply: We agree with the reviewer that there is still large uncertainty related to the understanding and characterization of non-CO₂ effects, as it is shown by the relatively large error bars in Figures 2, 3 and 4. We have reformulated the indicated paragraph in the Introduction accordingly:

“Although associated with large uncertainties, the understanding of non-CO₂ effects has improved over the years, allowing characterization of the relationship between atmospheric SLCF emissions and increase in radiative forcing (RF)

within acceptable intervals of confidence (see Lee et al.⁴ and Allen and co-workers^{5–8} for the relevant state-of-the-art).”

Reviewer #2:

Right now, for every \$100 that an airline spends to operate a flight, about \$20 goes to purchase the needed fuel, and about \$6 goes for the cost of purchasing and maintaining the airplane. A new medium-sized airplane costs about \$30 million. So that means that for every one of these airplanes in operation, something like \$100 million is being spent, somewhere, to build and operate the energy infrastructure needed to supply it with fuel. Right now that is a gigantic global network of oil wells, pipelines, ships, and refineries. If you were to instead spend those \$100 million on solar PV panels, the retail cost of which is about \$50 per square meter and which require virtually no money to operate, it means that you could cover 200 hectares of land. Wow! That’s a lot, occupying far more space than the plane itself. But it’s the way the world works.

Meanwhile, other stuff is happening. First, mainly because the cost of flying relative to people’s incomes has been falling, the amount people fly – and the investments into new planes and new energy infrastructure – has been growing, at something like 2.5% per year. If this trend continues, which airlines are planning on, the number of planes in the sky could triple by 2050. Second, most countries have decided to achieve net-zero emissions, effectively phasing out the use of fossil fuels across their economies. In Europe, the plan for the aviation sector, consistent with the net-zero economy-wide target, is to make sure that by 2065 all of the planes in operation are burning fuel derived from renewable sources, like solar power. To achieve the net-zero climate target they may also have to start requiring offsetting the non-CO₂ emissions from flying. We don’t know if these two planned trends – the growth in aviation volume and the move towards net-zero emissions – are compatible with each other. It probably depends on how much it costs to produce the renewable fuel and pay for the offsetting. If it costs about the same as it does to produce fossil kerosene, then the growth trend may continue. If it were to cost three times as much, then instead of \$20 going for fuel, airlines would have to spend \$60 for fuel and offsetting, pushing up their total costs by 40%. Given observed price elasticities, that would reduce demand for flying by about the same amount – 40% – meaning that instead of air traffic volume growing to 300% of its current level, it would grow to only 180%. If the cost of the fuel and offsetting were even higher, which would likely be the case if there were severe resource constraints, then the growth in aviation volume would be even less, perhaps even becoming negative.

The authors of this paper examine whether there would be severe resource constraints, going through a great deal of analysis to reach the conclusion that there would be, and that the decarbonization plans are inconsistent with the aviation growth plans. The problem is that where they actually compare what would be needed with what is available, they aren’t doing so for resources. Mainly they compare the electricity that would be needed for the fuels production to serve Europe with the electricity Europe currently produces. Electricity is not a fixed resource. It currently represents a rather small share of the energy system, since we use electricity for fairly small stuff, things like lighting, refrigeration, and computing. As we start using electricity for the big stuff, like driving cars and flying planes and heating our buildings, it is clear that investment flows will have to shift from fossil infrastructure to electricity production infrastructure, like those 200 hectares of solar panels that could be bought for each plane in the sky if we were to redirect investment.

Of course there are resource constraints that could actually constrain this needed growth in electricity production: water, land, materials, and labour. How these constraints play out depends, of course, depends on the technology used for generating electricity. If the promises people make of new nuclear reactors or cold fusion become reality by the second half of the century, then the resource needs will be quite small. Personally I am sceptical of this, and think it is more likely that we will get the electricity from solar PV, CSP, and wind power.

Previous work has shown that there clearly is enough land, assuming one uses land that is most suited, such as in the various deserts of the world. The authors of this paper suggest we would need about 4,160 TWh of electricity production per year. If you put that solar power in a place with a GHI of over 2,000 kWh / m², and assume a solar conversion efficiency of 20%, that means that you would need to cover a little over 10'000 square kilometres of land. That is less than 0.5% of the area of Saudi Arabia, a country that is going to have to figure out how to replace the likely collapse in oil export revenues.

Previous work has also shown that water is not a constraint. Thinking purely about energy production, Damerau et al. (Climatic Change, 2015) compared the water requirements for operating CSP plants (which relative to PV needs a lot of water), and found these to be less than half of the water needed to extract and refine oil with the same energy content. If we think about the water needing to be electrolyzed for the fuels production, we know that the current state-of-the air DAC technology extracts more than this from the air – even arid air in desert regions – simply as an unintended by-product of CO₂ capture.

Materials could be a constraint, although the authors do not examine this. Similarly, they do not examine whether labour availability could be a constraint. In either case, though, we need to think about the electricity needs of aviation in the context of the electricity needs of the entire economy. The decarbonization of road transport, for example, will require the switch to electric vehicles (which, one might be tempted to say, is clearly impossible, since the required volume of battery production for cars alone would exceed all the lithium ion batteries we are currently producing, for things like mobile phones and laptop computers), and this in turn will require vastly more electricity than what will be needed for the much smaller aviation sector. Really, the question is whether decarbonization of the economy will fly, where aviation represents one small piece of this. It is the overall decarbonization that will determine whether we push up against resource constraints. And those resource constraints will, partly, influence costs, which in turn will have a feedback on demand. Meanwhile, my understanding of the literature is that, currently, we do not view either labour supply or material reserves to be a major constraint. For example, the Transport chapter in the most recent IPCC report investigated the issue of material constraints for the electrification of ground transport – where many argue the issue is most acute – and did not find them to be a problem.

The authors of this paper convincingly show that what is needed to decarbonize aviation is breathtakingly large, if aviation demand continues to grow at the same rate it has grown before there were any efforts to decarbonize it, and costs were consistently declining. That is important to recognize. The modelling framework they have developed is impressive, and could be useful for estimating the effects of the planned decarbonization efforts on needed investment flows and materials usage. At the same time, one needs to recognize that there are huge error bars. For example, they assume the price of electricity will decline to about €50 per MWh by 2050, and then stay there. Others, such as Way et al. (Joule, 2022) suggest that even by 2030 the cost could well be below €10 per MWh.

While it pains me, I am going to recommend rejection of this paper. The work that has gone into it is impressive. But the way that they think about answering their question – whether we can decarbonize aviation – is fundamentally misleading.

Reply: In response to the reviewer's comments, we would like to start by thanking him/her for raising these important points and for the positive appreciations concerning our modelling framework. We regret the recommendation to reject our submission due to a potentially misleading way of answering our question.

While we agree that aviation is just one of the sectors of the economy to be decarbonized and that success or failure to do so will depend on developments in other sectors, we disagree with the hypothesis that sector-specific analyses (which ours could be considered to be) are misleading and must not be published. Especially, because our analysis of aviation is linked to the most important “external” factor, namely the dynamic development of the electricity sector, via prospective Life Cycle Assessment. And this linkage is performed using scenarios of global integrated assessment models, which do represent the development of the global economy under certain boundary conditions. Such sector specific studies represent incremental improvement and development to establish a better understanding of decarbonization efforts. These can then be further combined and integrated in global, economy-wide analyses (e.g., into integrated assessment modeling frameworks, which currently ignore non-CO₂ climate impacts of aviation) as the reviewer also seems to suggest. This is the way how science works, as far as we understand it. And this is also in line with the publication policy of *nature*, which has recently published several articles on the decarbonization of aviation (we now provide a new overview on recent literature in the Supporting Information).

In the following, we briefly explain how we use the issues raised by the reviewer to improve our manuscript:

- Concerning uncertainties, error bars and assumptions, as suggested also by reviewer 1, we have additionally performed a sensitivity analysis of the results to some key input parameters, e.g., cost of electricity, flight demand trajectories, etc. The results of these sensitivity analyses are presented in the Supplementary Information document. In a nutshell, our results are robust.
- Regarding potential constraints not taken into account – materials and labor – we added a remark on them in the *Climate-neutral European aviation and resources* section.

“On top of the potential resource demands we have quantified, there might be further constraints. Massive expansion of renewable power generation, hydrogen production and electric mobility – all key elements towards net-zero CO₂ emissions – are associated with substantial increases in demand for certain materials, which might exceed production capacities.”

Reviewer #3:

The work proposed by the authors is significant and interesting. It addresses an interesting topic which is prospective scenarios for climate-neutral aviation. The approach taken is relatively clear and the supplementary materials are significant and useful. Overall I think it is an interesting contribution for the community.

However, I have a few remarks given below but my main ones concern:

Comment 1: *the scope of the paper has to be more underlined especially in the abstract (maybe in the title?): European aviation*

Reply: We agree with the reviewer; we have modified the abstract to reflect the scope of our analysis and its focus on European aviation (see details below).

Comment 2: *the choice of considering only efuels makes it difficult to compare to other scenarios of the literature. Hence, I would underline that the conclusions of the paper are given with strong assumptions from the authors.*

Reply: As also suggested by Comment 2 of Reviewer 1, we have revised both the *Main* and *Discussion* sections to better reflect the scope of our analysis and to limit the conclusions to the mitigation options considered, i.e., offsetting via CO₂ removal by Direct Air Capture and permanent geological storage of CO₂, and the adoption of synthetic jet fuel produced from CO₂ captured from air and hydrogen synthesized through water electrolysis.

Comment 3: *the annual improvement of fleet efficiency (+0.6%) seems low to me, I would update that value with values of the proposed references. However, this will not change the conclusions regarding the need for traffic decrease.*

Reply: We thank the reviewer for the comment. We acknowledge that there might have been a misunderstanding on the assumed aircraft fleet efficiency. In our work, we have assumed an improvement in the engine efficiency of 0.6% per year; similarly, we have considered a yearly increase of 0.3% of each seating capacity and occupancy rate. This corresponds to a decrease in the fuel burn rate of 1.2% annually between 2018 and 2050 (in liters per passenger per km), and to an overall reduction of fuel use of 40% in the same period. Furthermore, we have analyzed the effect of a faster improvement in the fleet efficiency in the newly added sensitivity analysis (reported in the Supplementary Information), considering a 2% yearly decrease in fuel burn rate between 2018 and 2050, which corresponds to the aspirational goal of the International Civil Aviation Organization.

As we understand that the original manuscript was not clear in this respect, we have edited the following paragraph in the *Methods* section (*Aircrafts*):

“Expected annual improvements in terms of fuel efficiency, seating capacity and load factor between today and 2050 are aligned with those from the Destination 2050 roadmap report 2: 0.6%, 0.3% and 0.3%, respectively, to reach a cumulative improvement between 2020 and 2050 of 20%, 9% and 9%. **These improvements correspond to a total yearly decrease of the fuel burn rate (in liters per passenger per km) of 1.2% each year between 2018 and 2050.**”

Therefore, I would suggest to accept this paper after minor revisions.

Abstract

Comment 4: *The scope of the paper should be underlined: European aviation.*

Reply: We agree with the reviewer, and as mentioned in the reply to comment 1, we have edited the Abstract to underline the scope of the work:

“The **European** aviation sector must substantially reduce its climate impacts to reach net-zero goals. [...] Hence, the idea of a **European** climate-neutral aviation sector will fly only if air traffic decreases to reduce the scale of climate impacts to mitigate.”

1 – Introduction

Comment 5: *It would be interesting that at this point of the paper to underline the differences between biofuels and efuels as they imply different primary energy (biomass and low-carbon electricity) and do not have the same technological maturity. For biofuels and efuels I could suggest these references:*

*Zhao, X., Taheripour, F., Malina, R., Staples, M. D., & Tyner, W. E. (2021). Estimating induced land use change emissions for sustainable aviation biofuel pathways. *Science of the Total Environment*, 779, 146238.*

*Staples, M. D., Malina, R., Suresh, P., Hileman, J. I., & Barrett, S. R. (2018). Aviation CO2 emissions reductions from the use of alternative jet fuels. *Energy Policy*, 114, 342-354.*

Even if the paper focuses on efuels, most of the scenarios in the academic or institutional literature show the usage of biofuels before efuels.

Reply: We agree that our focus on electricity-based synthetic fuels needs to be emphasized and justified. We do so by adding “...and use of synthetic, **electricity-based** fuels.” in the *Introduction* section. Further, we add the following in the corresponding section of the main.

“While the small quantities of SAF used today are predominantly based on biomass (e.g., produced from used cooking oil), our focus is motivated by the fact that sustainable biomass resources (residues from forestry, agriculture, food) are limited and their utilization faces competition. Biomass-based SAF production could be scaled up using dedicated crops, but such feedstock often causes land use changes and associated, partially substantial climate impacts and further competes with land needed for food production.”

Comment 6: *Also regarding recent literature of prospective scenarios that include the climate impact of non-CO2 effects I think that these three articles would have their place in this introduction:*

*Planès, T., Delbecq, S., Pommier-Budinger, V., & Bénard, E. (2021). Simulation and evaluation of sustainable climate trajectories for aviation. *Journal of Environmental Management*, 295, 113079.*

*Grewe, V., Gangoli Rao, A., Grönstedt, T., Xisto, C., Linke, F., Melkert, J., ... & Dahlmann, K. (2021). Evaluating the climate impact of aviation emission scenarios towards the Paris agreement including COVID-19 effects. *Nature Communications*, 12(1), 1-10.*

Klöwer, M., Allen, M. R., Lee, D. S., Proud, S. R., Gallagher, L., & Skowron, A. (2021). Quantifying aviation's contribution to global warming. Environmental Research Letters, 16(10), 104027.

Nevertheless, there are many relevant references and I think it is really relevant to mention the report on Aviation's "missed" climate targets.

Reply: We thank the reviewer for the suggestion on articles worth citing in the Introduction. In fact, the articles by Grewe et al. and by Klöwer et al. are already cited in the Introduction. Furthermore, considering the various recent publications on the topics of non-CO₂ effects and the climate impact of aviation, we deemed it pertinent to present a table outlining the divergences and convergences between our research and the existing literature (including also the article by Planès et al. mentioned by the reviewer). The table has been added in the *Previous works* section under the *Methods* part of the manuscript.

Finally, we agree that the report *Missed Targets – A brief history of aviation climate targets*, which we included in the Intro is a very relevant citation.

2 – Main

Comment 7: *In this section the authors present three scenarios that they study. The choices they make seem relevant and for some original. Regarding climate objectives it would be worth to underline that the Paris Agreement aims to remain well below 2°C.*

Reply: Thank you for this comment. We modified the text accordingly (see *Main* section):

“...aiming at a global temperature increase **well** below 2°C...”

2.1 - Conventional jet fuel vs. Synthetic jet fuel

Comment 8: *It is worth mentioning that including the RF caused by DACCS is really original. From what I understand synthetic jet fuels only include efuels. It would be worth to mention that the scenarios presented consider that this technology is not yet mature and even 1% in 2026 as considered seems for me very ambitious.*

Reply: We agree with the reviewer about the ambitious target for synthetic jet fuel considered in our work. Besides adding a remark in the *Main* section on this aspect (see below), we have also performed a sensitivity analysis utilizing a lower target for the mandatory content of synthetic jet fuel in fuel blends, which is consistent with the sub-target proposed for these fuels by the EU ReFuel initiative.

“Following the EC-proposed ReFuelEU initiative, we assume that syn-jet fuel is initially blended with conventional jet fuel, with a volume percentage of 5% in 2030, 63% in 2050, and finally 100% in 2063 (i.e., +2.6% per year), **an arguably ambitious target because the technology exhibits still a relatively low technology**

readiness level, and scaling up the production in a cost-competitive way might be challenging.”

2.2 - Climate-neutral European aviation and resources

Comment 10: *It is assumed that the production of efuels are in Europe. Why not in a second part assume it is produced somewhere more appropriate? The least would be to underline that it is unfeasible at European scale.*

Reply: Indeed, we do explicitly model the production of efuels in Europe and contrast the resource requirements associated with efuel production (and climate impact mitigation) with European resource potentials in the *Climate-neutral European aviation and resources* section. It is precisely to underline that large-scale production of synthetic jet fuel (and climate change mitigation) for Europe cannot be achieved domestically.

However, assuming non-European efuels production would likely not alter our results and conclusions: REMIND projections place Europe at the forefront of electricity decarbonization in most scenarios. Hence, producing efuels elsewhere than in Europe, using grid electricity, is likely to require even more resources (as additional DACCS is needed to compensate for a more CO₂-intensive electricity). However, we expand the corresponding text in the manuscript to be more explicit regarding the implications of our results:

“Such massive demand for decarbonized electricity, land, and freshwater suggests that Europe's entire large-scale syn-jet fuel supply cannot be produced domestically. **At the same time, other regions like the Middle East, Australia, or South America might offer more promising opportunities to supply the required resources, in addition to satisfying their own domestic demand.**”

3 – Discussion

Comment 11: *Discussions are interesting but maybe lack of concrete quantified indicators regarding the amount of CDR or energy. This would help justify words like “major” “intensive”. Furthermore, I think it is important to mention that these conclusions rely on assumptions of the studied scenarios and pros of the aviation sector may underline that improvement in efficiencies are low (%0.6 per year) and that biofuels are not considered.*

Reply: We agree with the reviewer on the lack of quantitative information in the *Discussion* section. Following the reviewer’s comment, we have edited this section by adding specific quantitative information to reinforce our conclusions. Furthermore, as indicated in comment 2 of this reviewer and in comment 2 of reviewer 1, we have clarified in the Discussion that our conclusions are relevant to the mitigation options considered (i.e., CDR through DACCS and synthetic, electricity-based jet fuel), which may be combined with other technologies in the future. Finally, as explained in the reply to comment 3 of reviewer 1, the overall improvement in efficiencies correspond to 1.2%, and a larger improvement (2%) has been considered in the sensitivity analysis.

4 – Methods

Comment 12: *Generally, this section misses quantified examples and indicators. I encourage the authors to use more of the supplementary data in the text.*

Reply: As suggested in the previous comment, we have modified the *Discussion* section to add more quantitative results to support our qualitative observations. Further, we added quantified examples and indicators in the methods part – for example (table 10), the composition of global electricity generation in terms of technologies and resulting life-cycle GHG emissions of the electricity mixes over time; and (table 11) life-cycle emission factors syn-jet fuel production and combustion.

4.1 - General workflow

4.2 - Life-cycle environmental and cost modelling

Comment 13: *The authors use a life-cycle inventory database that is prospective which makes the work relatively original.*

Reply: We thank the reviewer for the positive comment and appreciation.

4.3 - European fleet scenarios and demand projection

4.4 – Aircrafts

Comment 14: *0.6 % is low compared to 2%-2.5% historic mentioned in the introduction. The Destination 2050 values are quite delicate to manipulate because they also introduce in 2035 “Improved technology (hydrogen) “ that contains energy efficiency also. In the approach chosen by the authors there is no coefficient for propulsive efficiency that is more significant than aerodynamics and weight. I think that improvement in efficiency should be refined to probably reach >1% as in Planès et al, Grewe et al, and Klower et al articles. Nevertheless, this should not change conclusions but provide more credibility to the work and results.
In addition, improvement in operations and load factors could be mentioned and discussed.*

Reply: We acknowledge that there might have been a misunderstanding on the assumed improvement in fleet efficiency. We refer the reviewer to our reply to comment 3 above, which addressed the points raised in this comment.

4.5 - Surface and flight emissions

4.6 - Radiative forcing and warming contribution of emissions

Comment 15: *Table 3 and 4 are interesting data and the approach chosen for non-CO2 effects based on Allen et al is clear. However, it would be interesting to compare with approach from Planès et al, Grewe et al, and Klower et al as there are no consensus yet on the approach.*

Reply: As the reviewer points out, recently there have been several publications on the topics of non-CO₂ effects and the climate impact of aviation, among which the ones cited in the comment above. Therefore, as explained in our reply to comment 6, we have added in the Supplementary Information a table outlining the divergences and convergences between our research and the existing literature (including also the articles mentioned by the reviewer). The table reports also on the approaches chosen in the various studies on how to treat non-CO₂ climate impacts. In our work, we followed the novel approach used by Klöwer et al., i.e., the linear-warming equivalent method. We consider a quantitative analysis – i.e., performing our analysis using different approaches for quantifying non-CO₂ effects – as beyond the scope of our current analysis, but as an interesting topic for a follow-up publication.

4.7 - Costs analysis

4.8 - Fuel supply

Comment 16: *Something compels me in this section. A single value is given for the content of CO₂ in a kg of syn-fuel. This depends strongly on the emission factor of electricity which (hopefully) should decrease with time. What electricity grid is considered? What about dedicated grid? Why is it not prospective as for the life-cycle inventory database?*

Reply: We think that there is a misunderstanding. While indeed the combustion emission factor of 3.14 kg CO₂ per kg (syn-)jet fuel remains constant over time (as determined by the fuel specifications), the life-cycle emission factor (production and combustion) of syn-fuel changes (decreases) over time. Which is indeed mainly an effect of decarbonization of electricity supply for electrolysis and DAC (and partly also thanks to an improvement of the electrolyzers and DAC over time). We provide these figures in table 11 in the section *Prospective life-cycle inventory database*.

5 - Direct air capture and CO₂ storage

6 - Prospective life-cycle inventory database

Comment 17: *I think I have my answer for 4.8 but this means I is not straightforward to understand and should be explained more thoroughly in 4.8.*

Reply: We agree that this might have been not transparent and hard to understand. We have explicitly added life-cycle emission factors of electricity generation and syn-jet fuel production and combustion in this section.

REVIEWER COMMENTS

Reviewer #1 (Remarks to the Author):

I appreciate the effort made by the authors to follow up on the review comments, in particular the sensitivity analysis which I'm sure was not a trivial effort. I have no additional comments at this stage.

Reviewer #2 (Remarks to the Author):

I want to thank the authors for making extensive, and to my mind really constructive edits to the paper. I also need to own the fact that in writing my first review, I got sidetracked by what I perceived as the authors' insensitivity to the way that the economy works. In doing so, I failed to comment on important aspects of the paper.

One of these aspects I raise now, the one remaining flaw in the paper, concerns the CO₂ intensity of future electricity supply. The authors assume various numbers, which are consistent with IPCC scenarios for the future energy sector (SSP2 for CRP 4.5 and 2.6). Those IPCC scenarios, however, did not take into account a switch to synthetic aviation fuel, as well as largely neglecting the rapid diffusion of electric road vehicles that has taken place since the SSPs were developed, and is now very likely to accelerate. For both of these reasons, the SSP 2 scenarios do not envision the kind of dramatic expansion of electricity supply that would be commensurate with the electrification of transportation (in the case of aviation through synthetic fuels production). So in other words, the SSP 2 envisions a completely different world for electricity production than the ones that this paper is consideration. Is it still valid to stick with the same assumptions of CO₂ emissions per kWh across these two worlds? I don't think so.

The electricity for synthetic fuel production represent a major new source of electricity demand. Clearly, the electricity is not going to simply come out of the existing grid, since that grid would crash. At the same time, the adoption of synthetic fuel is going to be driven by policy. It is only reasonable to expect the policy-makers crafting such legislation to consider the emissions associated with electricity supply, and if they are at all intelligent, to specify the LCA emissions that are allowable. Indeed this is exactly what European policy makers have done. Briefly: all synthetic aviation fuel qualifying for policy support needs to be produced using new sources of electricity, and these new sources have to be the lowest carbon available, such as solar PV, wind, and geothermal. Currently, the LCA emissions from these electricity sources stand at under 50 g / kWh, and this will fall towards zero as the energy used along the value chain shifts towards renewables. This contrasts strongly with the authors' assumption that the LCA emissions for electricity will still be at 130 g / kWh in 2050.

Anyway, all of this needs to be in the manuscript. At the very least, one needs to present a scenario in which one assumes that the new electricity production used to produce these fuels is coming from 100% renewable sources, consistent with the EU proposed rule and any assumption of intelligence among policy-makers. I would highly recommend abandoning the estimates for carbon intensity of electricity in the SSPs, as these storylines simply represent a very different world, and don't make sense in a world with so much new electricity demand.

I suspect that with this correction, the results will change substantially. I look forward to reviewing these.

Response to Reviewers

Dear reviewers,

We thank you for the constructive feedback and valuable comments – addressing them and revising the manuscript accordingly indeed improved the quality of the manuscript. The critical element of our revision – following the advice of reviewer 2 and the editor – is the following:

- **We now use the 2°C scenario as “default” instead of the 3.5°C scenario.** This scenario is – in terms of electricity supply – in line with the latest EU directives on renewable fuels of non-biological origin (from February 2023)¹², which aims at ensuring that hydrogen to be used for synthetic fuel production will be produced using electricity from *additional* renewable installations.
- We modify the text and the graphs accordingly, including the sensitivity analysis in the supporting information.

Below we list our point-by-point replies and actions. Besides, we have made minor text revisions to increase clarity and readability.

Reviewer #1:

I appreciate the effort made by the authors to follow up on the review comments, in particular the sensitivity analysis which I'm sure was not a trivial effort. I have no additional comments at this stage.

Response:

We thank the reviewer for the positive comment and the appreciation of our work. Based on the comment of reviewer #2 (below) and acknowledging the latest European directives regarding the electricity supply for hydrogen-based synthetic fuels, we have now considered the 2°C scenario as our default scenario instead of the 3.5°C scenario, as the 2°C degrees scenario complies with the relevant EU directives in terms of electricity supply. Therefore, the graphs showing the results of our analysis have (slightly) changed, but the main messages remain. Please see our response to reviewer #2 for more details.

Reviewer #2:

I want to thank the authors for making extensive, and to my mind really constructive edits to the paper. I also need to own the fact that in writing my first review, I got sidetracked by what

¹ Delegated regulation on Union methodology for RFNBOs. https://energy.ec.europa.eu/publications/delegated-regulation-union-methodology-rfnbos_en

² Delegated regulation for a minimum threshold for GHG savings of recycled carbon fuels and annex. https://energy.ec.europa.eu/publications/delegated-regulation-minimum-threshold-ghg-savings-recycled-carbon-fuels-and-annex_en

I perceived as the authors' insensitivity to the way that the economy works. In doing so, I failed to comment on important aspects of the paper.

One of these aspects I raise now, the one remaining flaw in the paper, concerns the CO₂ intensity of future electricity supply. The authors assume various numbers, which are consistent with IPCC scenarios for the future energy sector (SSP2 for CRP 4.5 and 2.6). Those IPCC scenarios, however, did not take into account a switch to synthetic aviation fuel, as well as largely neglecting the rapid diffusion of electric road vehicles that has taken place since the SSPs were developed, and is now very likely to accelerate. For both of these reasons, the SSP 2 scenarios do not envision the kind of dramatic expansion of electricity supply that would be commensurate with the electrification of transportation (in the case of aviation through synthetic fuels production). So in other words, the SSP 2 envisions a completely different world for electricity production than the ones that this paper is consideration. Is it still valid to stick with the same assumptions of CO₂ emissions per kWh across these two worlds? I don't think so.

The electricity for synthetic fuel production represent a major new source of electricity demand. Clearly, the electricity is not going to simply come out of the existing grid, since that grid would crash. At the same time, the adoption of synthetic fuel is going to be driven by policy. It is only reasonable to expect the policy-makers crafting such legislation to consider the emissions associated with electricity supply, and if they are at all intelligent, to specify the LCA emissions that are allowable. Indeed this is exactly what European policy makers have done. Briefly: all synthetic aviation fuel qualifying for policy support needs to be produced using new sources of electricity, and these new sources have to be the lowest carbon available, such as solar PV, wind, and geothermal. Currently, the LCA emissions from these electricity sources stand at under 50 g / kWh, and this will fall towards zero as the energy used along the value chain shifts towards renewables. This contrasts strongly with the authors' assumption that the LCA emissions for electricity will still be at 130 g / kWh in 2050.

Anyway, all of this needs to be in the manuscript. At the very least, one needs to present a scenario in which one assumes that the new electricity production used to produce these fuels is coming from 100% renewable sources, consistent with the EU proposed rule and any assumption of intelligence among policy-makers. I would highly recommend abandoning the estimates for carbon intensity of electricity in the SSPs, as these storylines simply represent a very different world, and don't make sense in a world with so much new electricity demand.

I suspect that with this correction, the results will change substantially. I look forward to reviewing these.

Response:

We are very grateful for these considerations, particularly the reviewer's willingness to provide constructive criticism. We consider the Reviewer's comments very valid, and we have correctly – we believe – reacted to them. Because indeed, our previous “default scenario” (the 3.5°C scenario) is not in line with recent EU legislation (February 2023) regarding electricity supply for synthetic fuel production and exhibits higher shares of fossil power generation than allowed by the EU directives. However, our 2°C scenario is indeed in line with these directives. Therefore, we use the 2°C scenario as our “default” and refer to the associated results in our discussion. The new choice of the reference scenario leads to substantially reduced costs of climate- and warming-neutral aviation. Still, regarding resource requirements, differences are minor; consequently, our main conclusions and messages hold.

The 2°C scenario complies with the EU directives, which aim at ensuring the water electrolysis for hydrogen production is mainly supplied with additional, dedicated renewable power from 2030 on, but it also allows for using grid electricity with a carbon intensity of below 65 g CO₂/kWh. We show that in Table 10 in the methods section of our manuscript: by 2030, the share of fossil fuels in power generation will be below 1%, and the associated life-cycle CO₂ emissions are 89 g CO₂/kWh. This higher value, compared to the EU directive, is because, unlike the EU directive which assigns zero GHG emissions to renewables (except for biomass) by excluding the emissions associated with the production of capital goods (e.g., wind turbines and PV panels), we perform a full scope Life Cycle Assessment including those emissions. The phase-out of fossil power generation in 2030 in our 2°C scenario ensures that only renewable power is used for hydrogen production. On top, it is also very likely *additional* renewable power (even considering the general trend of electrification) because the total EU electricity production in the 2°C scenario almost doubles compared to today by 2050 and increases practically threefold by 2100. In contrast, according to the more “business as usual” 3.5°C scenario, the EU electricity production in 2050 increases only by about 15% and in 2100 by 30% – which does not correspond to an electrification scenario. These absolute amounts of electricity generation were added in Table 10.

We have explained our new scenario choice in the manuscript and modified the text and graphs accordingly.

Specifically, we introduced the context of the EU directive in our introduction:

“Nevertheless, a recent delegated regulation of the European Parliament and Council establishes a minimum threshold for greenhouse gas emissions savings of synthetic fuels (equal to 70%) and requires such fuels to be produced almost exclusively with additional renewable electricity.”

We added the following in the main:

“In the following, we refer to the results for the 2° C scenario, which features an electricity mix that complies with the recent European delegated regulation for recycled carbon fuels including synthetic, electricity-based jet fuel (see Methods, Table 10, for electricity mix compositions). The regulation requires that renewable liquids are produced solely via additional renewable electricity or via grid electricity with a greenhouse gas emission intensity below ca. 65 g CO₂/kWh. Results for the 3.5° C scenario are presented under Extended Data Figures, reflecting a situation where low-carbon electricity is not available in sufficient quantity.”

We modified figures 2, 3, and 4 to show 2°C scenario results.

We added the following before Table 10:

“The share of electricity from fossil fuels is below 1% in the 2° C scenario in 2030 and thus in line with the latest EU directive on GHG emission savings of renewable liquid and gaseous transport fuels of non-biological origin and from recycled carbon fuels. An important difference between our rigorous LCA approach and the EU directive is the fact that while we do quantify emissions associated with the production of capital goods (e.g., wind turbines and PV panels), the EU directive does not and instead associates electricity from wind, solar, geothermal and hydro power with zero GHG emissions.”

We added the following after Table 11:

“The figures provided here must not be directly compared with the recent EU directive on GHG emission savings of renewable liquid and gaseous transport fuels of non-biological origin and from recycled carbon fuels, since the GHG emission reduction threshold of 70% required

according to the directive is based on zero GHG emissions from non-biomass renewable power, while we quantify complete life cycle emissions.”

Further, minor text edits are documented per track change.

REVIEWER COMMENTS

Reviewer #2 (Remarks to the Author):

I congratulate the authors on doing a healthy amount of additional analysis, responding to my earlier comments. I now have very few concerns. They are as follows:

Line 122: The manuscript says that it bases the LCA analysis, including geological storage capacity, land, and freshwater use on Cox et al. 2018 (ref. 49) and Becattini et al. 2021 (ref 50). Neither of these papers present any LCA analyses for these issues for RFNBOs.

Figures 2 and 3: in the growth scenario, pure DACCS offsetting, DACCS is at 1 billion tons by 2060 – 70, and rises to 1.5 billion tons CO₂ by 2100. In the stabilization scenario, DACCS stays at about 0.25 billion tons CO₂. In the growth scenario for RFNBOs, DACCS is at 0.5 billion tons CO₂ by 2060 – 70, and rises to 0.8 billion tons by 2100. In the stabilization scenario, DACCS stays below 0.1 billion tons. Compare what would be required in negative emissions globally: scenarios consistent with 1.5 – 2°C warming suggest up to 7 billion tons CO₂ NETS by the time society reaches net zero, and up to 15 billion tons net negative by 2100. It would be important to put this into context. Yes, under a growth scenario, the DACCS requirements for European aviation would represent a non-trivial contribution to overall global negative emissions requirements. Under the scenario with continued fossil fuel use and full DACCS offsetting, it would account for close to 10% of total global negative emissions, while in the RFNBO scenario it would account for roughly 5%.

Energy use: the article suggests that electricity for hydrogen production between 2050 and 2100 would be 1.3 times current EU electricity use. OK. And the use of fossil jet fuel would 100% lower. I am not sure how the calculations for land and freshwater use are derived. Lewt's go in assuming that aviation fuel markets will continue to be global: so the starting point is the assumption that fuels will be produced in areas with exceptionally good and inexpensive renewable energy potentials, such as the Middle East, Australia, or South America.

How do the authors arrive at 20 g CO₂ / kWh in 2100? I could not find support in the Methods section. If indeed the EU has achieved net zero by 2050, including zero in the energy sector, then presumably all of the inputs into electricity production (and everything else, including the Airbus factory in Toulouse) from 2050 onward, to the extent they take place in Europe, would also result in zero emissions. Presumably Europe would put in place regulations ensuring that imported fuels would have the same carbon content, i.e. 0, of fuels produced in Europe. And one should not forget that most countries now do have net zero targets for 2050.

The paper claims high amounts of freshwater use associated with RFNBO production. I cannot find the sources to justify this result, and also to put this into context of total freshwater use. I recall that Damerou et al. (2015) (<https://link.springer.com/article/10.1007/s10584-015-1345-y>) compared water use for fossil fuel and renewable electricity production in the MENA region, presumably one place where RFNBO production would be likely to take place. Measured against energy output, they found water demand to be far lower by 2100 in a scenario dominated by renewable electricity exports, compared to one dominated by fossil fuel exports. The driving factor is the increasing use of unconventional oil extraction, which is water intensive. Even accounting for the inefficiencies of converting electricity into RFNBOs, the results would suggest that the switch to RFNBOs could potentially result in a water savings. So to make any claims about freshwater availability being a constraint, the authors need to clarify the methods used to obtain the water use estimates, and compare these with water use

estimates associated with continued fossil fuel use, and also with water use requirements in production areas outside of the energy sector.

The paper also claims high land area requirements. I could not actually find the details on this, but I assume that the authors assume RFNBO production in regions with high solar radiation and wind speeds. To make any claims about land-use being a constraint, it is essential to compare the land utilization in these areas with total land availability.

The sentence starting with line 350 "Thus, reducing air-traffic demand is a good short-to-mid-term solution" is unsupported. The authors have not analysed the political, social, and economic factors associated with steps required to reduce air traffic demand. Similarly, the authors have not analysed the economic burdens of RFNBO utilisation under a growth scenario: we know this will require large amounts of investments, but this would need to be compared with existing levels. Furthermore, these investments would only be made if the air travel they enabled led to a welfare increase; it would be air passengers who would decide whether or not to fly, at the higher cost. I would suggest something like "From a purely physical standpoint, reducing air-traffic demand is an attractive short- to mid-term solution." The sentence after also needs changing. We simply do not know if reducing demand reduces the environmental and economic effort. If, for example, aviation were replaced with rail or shipping transport, both of which are far more expensive per passenger kilometre and require far greater amounts of infrastructure and equipment, then reducing aviation demand would in fact lead to greater environmental and economic effort. What might lead to lower effort would be a reduction in demand for transport. But even here, we don't know what other effects this might incur. As a trival example, residents of Berlin can opt, instead of flying to tropical beaches, to visit the indoor Tropical Islands facility. Do we know that heating this facility have a lower environmental footprint that flying the same number of people to Mallorca, where they can then spend their time outdoors? The final sentence before the Methods section relies on the other two, and hence also needs changing. Similarly, the final sentence of the Abstract needs changing to reflect the lack of analysis into the impacts associated with demand reduction. The results in the paper do not support the statement that "European climate-neutral aviation will fly only if air traffic demand decreases ..."

Response to Reviewer

Dear reviewer,

We appreciate the quick handling of our second manuscript revision and thank you for providing us with additional review comments. As demonstrated in our previous two revisions, we highly value constructive feedback that helps us improve the quality of our work. In this third revision, we have taken into consideration the comments from Reviewer #2 in a constructive and thorough manner (please find our point-by-point responses below).

Reviewer #2:

Comment:

Line 122: The manuscript says that it bases the LCA analysis, including geological storage capacity, land, and freshwater use on Cox et al. 2018 (ref. 49) and Becattini et al. 2021 (ref 50). Neither of these papers present any LCA analyses for these issues for RFNBOs.

Response:

We agree that our original wording here might have been misleading, and we have revised it, as requested. The LCA for these issues (i.e., incl. geological storage capacity, land, and freshwater use) for RFNBOs (renewable liquid and gaseous fuels of non-biological origin) is indeed one of the novelties of this manuscript. We developed and applied a new aircraft-synfuel-cost-LCA model to perform this LCA, which builds upon the two references mentioned.

We have resolved the issue by rephrasing the fragment mentioned by the reviewer as follows: “...are quantified. To this end, we develop and apply a new life-cycle environmental and cost assessment model for aircraft and electricity-based syn-jet fuel, which is described in the Methods section and builds upon earlier work of ours^{50,51}.”

Comment:

Figures 2 and 3: in the growth scenario, pure DACCS offsetting, DACCS is at 1 billion tons by 2060 – 70, and rises to 1.5 billion tons CO₂ by 2100. In the stabilization scenario, DACCS stays at about 0.25 billion tons CO₂. In the growth scenario for RFNBOs, DACCS is at 0.5 billion tons CO₂ by 2060 – 70, and rises to 0.8 billion tons by 2100. In the stabilization scenario, DACCS stays below 0.1 billion tons. Compare what would be required in negative emissions globally: scenarios consistent with 1.5 – 2°C warming suggest up to 7 billion tons CO₂ NETS by the time society reaches net zero, and up to 15 billion tons net negative by 2100. It would be important to put this into context. Yes, under a growth scenario, the DACCS requirements for European aviation would represent a non-trivial contribution to overall global negative emissions requirements. Under the scenario with continued fossil fuel use and full DACCS offsetting, it would account for close to 10% of total global negative emissions, while in the RFNBO scenario it would account for roughly 5%.

Response:

Thank you for these considerations – we are happy to see that our results are clear and stimulating enough to be put into such a context by the readers, even without explicitly providing it. We follow your suggestion and now add such context in the revised manuscript explicitly, by expanding the discussion as follows:

“For climate-neutrality, 1 and 1.7 Gt CO₂ must be removed annually between 2050 and 2100 for the syn- and (fossil) jet fuel options under a growing demand trajectory in a 2° C climate scenario, respectively (and 20% more in a 3.5° C climate scenario). [...]. It is worth noting that compared to the overall CDR requirements in the ensemble of IPCC’s 2° C scenarios, in which between close to zero and up to 5 Gt CO₂ in 2050 and about 5-15 Gt CO₂ in 2100 need to be removed, the amounts needed for a climate-neutral European aviation under the growing demand trajectory are substantial.”

Comment:

Energy use: the article suggests that electricity for hydrogen production between 2050 and 2100 would be 1.3 times current EU electricity use. OK. And the use of fossil jet fuel would 100% lower. I am not sure how the calculations for land and freshwater use are derived.

Response:

We agree that some additional explanation regarding land and freshwater use should be provided, as requested. Therefore, we have added the following at the beginning of the section “Resource impact modeling for European fleet scenarios”:

“The prospective life-cycle inventory database represents the basis for our quantification of freshwater abstraction and land use presented in Figure 4. Freshwater abstraction requirements represent the uncharacterized (i.e., not considering site-specific scarcity) cumulative flows of freshwater withdrawal (i.e., not considering the fate of the freshwater) in terms of volume in each of our scenarios. Land use represents the uncharacterized (i.e., not considering site-specific land quality) cumulative occupied area of urban and agricultural land over time (i.e., m² over a year) in each of our scenarios.”

Comment:

Let’s go in assuming that aviation fuel markets will continue to be global: so the starting point is the assumption that fuels will be produced in areas with exceptionally good and inexpensive renewable energy potentials, such as the Middle East, Australia, or South America.

How do the authors arrive at 20 g CO₂ / kWh in 2100? I could not find support in the Methods section. If indeed the EU has achieved net zero by 2050, including zero in the energy sector, then presumably all of the inputs into electricity production (and everything else, including the Airbus factory in Toulouse) from 2050 onward, to the extent they take place in Europe, would also result in zero emissions. Presumably Europe would put in place regulations ensuring that imported fuels would have the same carbon content, i.e. 0, of fuels produced in Europe. And one should not forget that most countries now do have net zero targets for 2050.

Response:

The 20 g CO₂ per kWh (as CO₂ emission intensity of electricity supply) are a result of life-cycle-based CO₂ intensities of individual power generation technologies multiplied by their market shares as provided in Table 10. These CO₂ intensities of individual technologies are derived by modifying the current life-cycle inventories of the ecoinvent LCA database¹ and applying the *premise* framework² to represent future developments in key economic sectors in line with our two IAM scenarios; this is explained in detail in the “Prospective life-cycle inventory database” section and in the references referred to therein. As a result, these emission intensities are (substantially) lower than today, but not equal to zero – as “global net-zero” is not the underlying storyline of the IAM scenarios we build upon.

In our opinion, the assumption that the net-zero goal will be achieved on a global level and thus LCA (for quantifying climate impacts) will be obsolete at a certain point in time is – from the current perspective and considering recent developments in terms of climate policy and emission reductions – way too much of a simplification and cannot be justified. Such an assumption would just be based on normative, political goals, but not scientific evidence. This is our opinion for several reasons: First, there is evidence that announced net-zero or carbon neutrality goals are often ambiguous – in terms of emission scope (CO₂ or GHG emissions), regarding definition and quantification of residual emissions, in terms of geographical scope (territorial emissions only or not), and regarding emission offsetting and CDR³⁻⁶. Second, these goals rely on large amounts of mostly forest and soil-based CDR, which “are reversible, prone to disturbances, and limited in the long-term owing to saturation”⁷. Third, even these ambiguous targets only cover less than 20% of today’s global CO₂ emissions³. And finally, because the actual trend in global GHG emissions, and in general internationally implemented policies and Nationally Determined Contributions are currently nowhere close to GHG emissions trajectories in line with limiting global warming to two degrees (maybe except for the EU)⁸.

Further, regarding European regulations on carbon intensities of synthetic fuels: As ambitious and appreciated they are – they do not represent carbon intensities equal to zero from an LCA perspective, since they assign zero GHG emissions to renewable power generation (exception: biomass-based) by not taking into account emissions associated with the production of capital goods (e.g., photovoltaic panels and wind turbines) and by ignoring GHG emissions emitted from reservoirs of hydropower plants^{9,10}. We did explain that in our previous reply already and have highlighted this difference in the methods section of the manuscript.

Finally, it is acknowledged that there is a lack of clarity surrounding the concept of net zero and its implications, especially when considering not only emissions of long-lived climate forcers (LLCF) as CO₂, but also those of short-lived climate forcers (SLCF), which are of crucial importance for aviation. A recent, authoritative review by Allen et al.¹¹ argues that a better understanding of this concept is necessary to effectively address climate change. The authors state that “It has become almost an article of faith that achieving net zero CO₂ emissions, or net zero CO_{2-e100} emissions, is necessary to meet our climate goals. Although this may be a helpful simplification for motivating climate policies today, it is not entirely rigorous [...] Net zero CO_{2-e100} emissions in the second half of this century may or may not be required to meet the goals of the Paris Agreement depending on the mix of LLCFs and SLCFs in the emissions scenario.”

For all these reasons, we advocate for and insist on our approach for quantifying GHG emissions, namely a consistent coupling of established IAM scenarios and prospective LCA. Accordingly, no changes to the manuscript have been made in response to this comment of the Reviewer.

Comment:

The paper claims high amounts of freshwater use associated with RFNBO production. I cannot find the sources to justify this result, and also to put this into context of total freshwater use. I recall that Damerau et al. (2015) (<https://link.springer.com/article/10.1007/s10584-015-1345-y>) compared water use for fossil fuel and renewable electricity production in the MENA region, presumably one place where RFNBO production would be likely to take place. Measured against energy output, they found water demand to be far lower by 2100 in a scenario dominated by renewable electricity exports, compared to one dominated by fossil fuel exports. The driving factor is the increasing use of unconventional oil extraction, which is water intensive. Even accounting for the inefficiencies of converting electricity into RFNBOs, the results would

suggest that the switch to RFNBOs could potentially result in a water savings. So to make any claims about freshwater availability being a constraint, the authors need to clarify the methods used to obtain the water use estimates, and compare these with water use estimates associated with continued fossil fuel use, and also with water use requirements in production areas outside of the energy sector.

The paper also claims high land area requirements. I could not actually find the details on this, but I assume that the authors assume RFNBO production in regions with high solar radiation and wind speeds. To make any claims about land-use being a constraint, it is essential to compare the land utilization in these areas with total land availability.

Response:

Thank you for signalling this publication. And we agree, the methods section lacked some details regarding quantification of natural resource use. We have added an explanation on how we quantify both freshwater and land use and what these metrics represent in the section “Resource impact modeling for European fleet scenarios”, as requested.

Beyond that: If we interpret this comment correctly, there might be a misunderstanding, which triggered this comment of the Reviewer. We try to clarify: In our manuscript, results are quantified for synthetic jet fuel production in Europe, without specifying locations more precisely, i.e., using generic costs representative for Europe and the average European electricity mix for water electrolysis. This is explained in the section “Prospective life-cycle inventory database” by the fragment “...As shown in Ref⁹⁵, the electricity sector is most influential in the context of prospective LCA in general and even more so in our analysis, as the contributions from European electricity supply dominate the release of GHG associated with syn-jet fuel production and DACCS.” To make this even more explicit, in the revised manuscript we have added the fragment “...at generic European locations” in the section “Conventional jet fuel vs. Synthetic jet fuel”.

Based on our results for resource needs, which we indeed put into the context of such resource consumptions in Europe (e.g., agricultural land use in EU-2018 and overall freshwater consumption in the EU-28) in Figure 4, we argue, page 11 of the manuscript, that “Such massive demand for decarbonized electricity, land, and freshwater suggests that Europe's entire large-scale syn-jet fuel supply may not be produced domestically. At the same time, other regions like the Middle East, Australia, or South America might offer more promising opportunities to supply the required resources (especially land and renewable electricity) – provided they satisfy their own domestic demand.”. This paragraph was present at the time of the initial submission.

Comment:

The sentence starting with line 350 “Thus, reducing air-traffic demand is a good short-to-mid-term solution” is unsupported. The authors have not analysed the political, social, and economic factors associated with steps required to reduce air traffic demand. Similarly, the authors have not analysed the economic burdens of RFNBO utilisation under a growth scenario: we know this will require large amounts of investments, but this would need to be compared with existing levels. Furthermore, these investments would only be made if the air travel they enabled led to a welfare increase; it would be air passengers who would decide whether or not to fly, at the higher cost. I would suggest something like “From a purely physical standpoint, reducing air-traffic demand is an attractive short- to mid-term solution.”

Response:

We agree and have modified this sentence according to your suggestion, as requested.

Comment:

The sentence after also needs changing. We simply do not know if reducing demand reduces the environmental and economic effort. If, for example, aviation were replaced with rail or shipping transport, both of which are far more expensive per passenger kilometre and require far greater amounts of infrastructure and equipment, then reducing aviation demand would in fact lead to greater environmental and economic effort. What might lead to lower effort would be a reduction in demand for transport. But even here, we don't know what other effects this might incur. As a trival example, residents of Berlin can opt, instead of flying to tropical beaches, to visit the indoor Tropical Islands facility. Do we know that heating this facility have a lower environmental footprint that flying the same number of people to Mallorca, where they can then spend their time outdoors? The final sentence before the Methods section relies on the other two, and hence also needs changing.

Response:

We respectfully disagree with the Reviewer. The sentence “It drastically reduces the scale of the environmental and economic effort needed to limit the impact of aviation on the climate.” is correct as it stands, since it refers to *the impact of aviation on the climate*. Aviation only, without any substitution or rebound effects. Those are out of and far beyond the scope of our analysis. There is no doubt that a demand reduction in aviation reduces environmental and economic efforts related to limiting climate impacts of the aviation sector as such.

Regarding modal shifts: Air travel is the transport mode with the highest climate impacts among realistic alternatives (see e.g., <https://pubs.acs.org/doi/10.1021/es4003718>). Thus, substituting air travel with trains, passenger vehicles or buses in our stationary and declining demand scenarios would still reduce transport related climate impacts.

However, we do agree that quantifying rebound effects would be interesting. We clarify the scope of our analysis and the need for further analysis about rebound effects by adding the following paragraph to the “main”: “Note that we do not consider any potential rebound effects associated with stationary or declining demand – people not spending money on flight tickets might spend it on other goods or services associated with other different climate impacts. However, examining such implications would require different models and approaches that are beyond the scope of this study. Additionally, we have not accounted for any potential shifts in transportation modes. It is however worth noting, that previous research has shown that air travel has by far the most significant climate impacts among all passenger transportation modes¹².”

Comment:

Similarly, the final sentence of the Abstract needs changing to reflect the lack of analysis into the impacts associated with demand reduction. The results in the paper do not support the statement that “European climate-neutral aviation will fly only if air traffic demand decreases ...”

Response:

We respectfully disagree with the reviewer. The abstract reflects the main outcome of our analysis, i.e., achieving a climate-neutral aviation sector in Europe would require substantial amounts of economic and natural resources. Therefore, we stand by our conclusion that a reduction in air traffic demand is a necessary step in achieving climate neutrality in the European aviation sector.

Our work did not include a geospatial analysis assessing the accessibility of regions that could provide the necessary resources, such as electricity, land, freshwater, CO2 storage volumes. Such an analysis could be a valuable contribution to the field, but it is far beyond the scope of

the analysis reported in this work, which can certainly serve as a basis for this further follow-up research.

1. Wernet, G., Bauer, C., Steubing, B., Reinhard, J., Moreno-Ruiz, E., and Weidema, B. (2016). The ecoinvent database version 3 (part I): overview and methodology. *Int. J. Life Cycle Assess.* *21*, 1218–1230.
2. Sacchi, R., Terlouw, T., Siala, K., Dirnaichner, A., Bauer, C., Cox, B., Mutel, C., Daioglou, V., and Luderer, G. (2022). PRospective EnvironMental Impact asSEment (premise): A streamlined approach to producing databases for prospective life cycle assessment using integrated assessment models. *Renew. Sustain. Energy Rev.* *160*, 112311. [10.1016/j.rser.2022.112311](https://doi.org/10.1016/j.rser.2022.112311).
3. Meinshausen, M., Lewis, J., McGlade, C., Gütschow, J., Nicholls, Z., Burdon, R., Cozzi, L., and Hackmann, B. (2022). Realization of Paris Agreement pledges may limit warming just below 2 °C. *Nat.* *2022* 6047905 *604*, 304–309. [10.1038/s41586-022-04553-z](https://doi.org/10.1038/s41586-022-04553-z).
4. Hale, T., Smith, S.M., Black, R., Cullen, K., Fay, B., Lang, J., and Mahmood, S. (2022). Assessing the rapidly-emerging landscape of net zero targets. *Clim. Policy* *22*, 18–29. [10.1080/14693062.2021.2013155/SUPPL_FILE/TCPO_A_2013155_SM5962.DOCX](https://doi.org/10.1080/14693062.2021.2013155/SUPPL_FILE/TCPO_A_2013155_SM5962.DOCX).
5. Buck, H.J., Carton, W., Lund, J.F., and Markusson, N. (2023). Why residual emissions matter right now. *Nat. Clim. Chang.* *2023* *134* *13*, 351–358. [10.1038/s41558-022-01592-2](https://doi.org/10.1038/s41558-022-01592-2).
6. Rogelj, J., Geden, O., Cowie, A., and Reisinger, A. (2021). Net-zero emissions targets are vague: three ways to fix. *Nat.* *2021* 5917850 *591*, 365–368. [10.1038/d41586-021-00662-3](https://doi.org/10.1038/d41586-021-00662-3).
7. Smith, H.B., Vaughan, N.E., and Forster, J. (2022). Long-term national climate strategies bet on forests and soils to reach net-zero. *Commun. Earth Environ.* *2022* *31* *3*, 1–12. [10.1038/s43247-022-00636-x](https://doi.org/10.1038/s43247-022-00636-x).
8. IPCC (2022). Synthesis Report of the Sixth Assessment Report.
9. Scherer, L., and Pfister, S. (2016). Hydropower’s Biogenic Carbon Footprint. *PLoS One* *11*, e0161947. [10.1371/JOURNAL.PONE.0161947](https://doi.org/10.1371/JOURNAL.PONE.0161947).
10. Hertwich, E.G. (2013). Addressing biogenic greenhouse gas emissions from hydropower in LCA. *Environ. Sci. Technol.* *47*, 9604–9611. [10.1021/ES401820P/SUPPL_FILE/ES401820P_SI_002.XLSX](https://doi.org/10.1021/ES401820P/SUPPL_FILE/ES401820P_SI_002.XLSX).
11. Allen, M.R., Friedlingstein, P., Girardin, C.A.J., Jenkins, S., Malhi, Y., Mitchell-Larson, E., Peters, G.P., and Rajamani, L. (2022). Annual Review of Environment and Resources Net Zero: Science, Origins, and Implications. [10.1146/annurev-environ-112320](https://doi.org/10.1146/annurev-environ-112320).
12. Borken-Kleefeld, J., Fuglestvedt, J., and Berntsen, T. (2013). Mode, load, and specific climate impact from passenger trips. *Environ. Sci. Technol.* *47*, 7608–7614. [10.1021/ES4003718/ASSET/IMAGES/LARGE/ES-2013-003718_0003.JPEG](https://doi.org/10.1021/ES4003718/ASSET/IMAGES/LARGE/ES-2013-003718_0003.JPEG).

REVIEWERS' COMMENTS

Reviewer #2 (Remarks to the Author):

The authors have made some useful changes. And yet, I am still concerned that some of the key results and findings, if taken out of context, could be misleading.

First, it is the assumption within the article that the production of fuels for the European aviation sector will take place within Europe, using electricity generated within Europe. While the authors have developed CO₂ emissions scenarios that are independent of this, the assumption still finds its way into the conclusions in terms of the estimates for land-use and freshwater use. I could not find it specifically, but I could only assume that the freshwater results were based primarily on the existing use of hydropower and biomass. Now, the assumption of European production goes unstated until the point where the authors acknowledge it by suggesting that such fuels may be imported from other countries. I would argue, and most experts I know would argue, that it is more likely than not that European aviation fuels will continue to be imported, whether fossil or synthetic. Ideally, this manuscript would do its calculations for land and water use based on that assumption. Or, alternatively, they could provide alternative results based on different assumptions: domestic European production using the current renewable electricity supply mix; production in low-cost places using solar; production in low-cost places using wind. This would entail a lot of additional analysis. I would be willing to live with simply more clearly acknowledging, both early in the manuscript and at the place where the water and land use results are described, that this rests on the assumption of European production using the current renewable generation mix, and that sourcing fuels from alternative places with high wind and solar resources would likely result in lower land and water requirements. For example, land requirements would be zero in the case of manufacturing using offshore wind energy.

Second, I disagree with considering only a scenario that simultaneously seeks to have the world achieve early net zero emissions, and assumes that LCA emissions from electricity production will remain positive. We are talking about a 50 year time lag here, between 2050 (when most countries have promised net zero emissions) and 2100 (when the authors assume there will be emissions associated with the production of new electricity-generation emissions). That is longer than the investment lifetimes of any of the equipment used in the manufacturing. Even if net zero gets reached in 2070, it is still a 30 year lag. Let's assume that China is the place where PV panels used for fuel production are manufactured, and that China reaches net zero emissions by 2070. So that means that starting in 2070, adding additional PV capacity will have zero effect on emissions. I would propose that by 2100, virtually none of the PV capacity installed prior to 2070 will still be operational. And that means that the LCA emissions would by then be 0. I would request the authors to consider a scenario in which LCA emissions do decline to 0 by 2100, or before. Or, at the very least, acknowledge the absence of such a scenario as a potential weakness in the paper.

Third, I agree with the authors' statement that aviation produces more climate impacts than other forms of travel. Clearly this is relevant when examining substitution effects (these are substitution effects, not rebound effects). But their article is examining the consequences of eliminating those climate impacts. And so at least in terms of economic issues, one needs to examine the economics. The simple fact is that aviation is the least expensive way to travel distances over a couple hundred kilometres, because of the very low infrastructure required. To make aviation climate neutral, fundamentally, will take a lot of infrastructure. So ultimately the question is whether one would need more infrastructure (at higher investment costs, and land use, and all that) with flying climate neutral, or travelling another way that is also climate neutral. This is why the conclusions reached in the final paragraph of the main manuscript (lines 355 - 364) need to be crafted carefully. From a physical standpoint, reducing aviation demand is attractive; we don't know if it is good until we compare the infrastructure effects that come with substitution. We also don't know if reducing aviation demand reduces the environmental and economic consequences, because (a) we are comparing two worlds

that are both climate neutral, and (b) we haven't examined the environmental and economic consequences of the substitution that is likely to occur.

Finally, the sentence in the abstract containing "climate-neutral aviation will fly only if air traffic is reduced" will, simply put and from a scientific perspective, not fly. The conclusions simply do not support it. The paper finds that huge amounts of infrastructure are required. Yes. Will society be willing to build this infrastructure? Nobody knows. We build huge infrastructure all the time in order to achieve things we want to achieve. I suspect that if people really want to fly, and governments are clear that flying needs to be climate neutral, then the infrastructure will be built, and the cost of building it will be reflected in airline ticket prices. That's how the economy works. I tend to think this is the more likely outcome than governments' somehow restricting people's ability to fly, taking into account political, social, and economic factors.

Response to Reviewer

Dear reviewer,

Again, we thank you for your time spent on our manuscript and your comments. We address these point by point below.

Reviewer #2:

Comment:

The authors have made some useful changes. And yet, I am still concerned that some of the key results and findings, if taken out of context, could be misleading.

First, it is the assumption within the article that the production of fuels for the European aviation sector will take place within Europe, using electricity generated within Europe. While the authors have developed CO₂ emissions scenarios that are independent of this, the assumption still finds its way into the conclusions in terms of the estimates for land-use and freshwater use. I could not find it specifically, but I could only assume that the freshwater results were based primarily on the existing use of hydropower and biomass. Now, the assumption of European production goes unstated until the point where the authors acknowledge it by suggesting that such fuels may be imported from other countries. I would argue, and most experts I know would argue, that it is more likely than not that European aviation fuels will continue to be imported, whether fossil or synthetic. Ideally, this manuscript would do its calculations for land and water use based on that assumption. Or, alternatively, they could provide alternative results based on different assumptions: domestic European production using the current renewable electricity supply mix; production in low-cost places using solar; production in low-cost places using wind. This would entail a lot of additional analysis. I would be willing to live with simply more clearly acknowledging, both early in the manuscript and at the place where the water and land use results are described, that this rests on the assumption of European production using the current renewable generation mix, and that sourcing fuels from alternative places with high wind and solar resources would likely result in lower land and water requirements. For example, land requirements would be zero in the case of manufacturing using offshore wind energy.

Response:

We agree with the reviewer that our default assumption that synthetic jet fuels are produced in Europe is an important one; in case synthetic jet fuels were produced elsewhere, the resource requirements would be different, either larger or smaller, or partly larger and partly smaller, depending on a set of assumptions different than the ones we make. Therefore, we add a sentence in the section “Climate-neutral European aviation and resources” to acknowledge this and to strengthen the point. We also add the assumption of synthetic jet fuel production in Europe in the methods section where we explain how we quantify resource demand (“Resource impact modeling for European fleet scenarios”).

However, we would like to clarify what we feel are a few misunderstandings:

- 1) The assumption of European production DOES NOT go “*unstated until the point where the authors acknowledge it by suggesting that such fuels may be imported from other countries*”, as the reviewer argues. On the contrary, we provide this information already in the section “Conventional jet fuel vs. Synthetic jet fuel”, when we say: “*The syn-jet fuel is produced through Fischer-Tropsch synthesis, fed by hydrogen (H₂) from water electrolysis and carbon monoxide (CO) from the reverse water-gas shift reaction using CO₂ from DAC at generic European locations*”.
- 2) Our results DO NOT rest on the assumption of European production using the *current* renewable generation mix. The entire sub-section “Prospective life-cycle inventory database” in the methods section explains in detail our prospective LCA approach. Most importantly, in our analysis, (i) we modify shares of renewable power generation over time in line with specific 2 degrees scenarios from the integrated assessment model REMIND; (ii) we also adjust life cycle inventories of renewable power generation technologies to represent their future improvement; and (iii) we modify the entire background inventory data to represent the expected transformation of the global economy – which leads for example to reduced environmental impacts of the production of steel used for wind turbine manufacturing.
- 3) No single technology will ever be produced with zero resource requirements, as we employ a life-cycle assessment approach. Direct land use of offshore wind power might be zero, but wind turbine production and end-of-life treatment will still occupy land, besides having a non-negligible footprint at sea.

Comment:

Second, I disagree with considering only a scenario that simultaneously seeks to have the world achieve early net zero emissions, and assumes that LCA emissions from electricity production will remain positive. We are talking about a 50 year time lag here, between 2050 (when most countries have promised net zero emissions) and 2100 (when the authors assume there will be emissions associated with the production of new electricity-generation emissions. That is longer than the investment lifetimes of any of the equipment used in the manufacturing. Even if net zero gets reached in 2070, it is still a 30 year lag. Let's assume that China is the place where PV panels used for fuel production are manufactured, and that China reaches net zero emissions by 2070. So that means that starting in 2070, adding additional PV capacity will have zero effect on emissions. I would propose that by 2100, virtually none of the PV capacity installed prior to 2070 will still be operational. And that means that the LCA emissions would by then be 0. I would request the authors to consider a scenario in which LCA emissions do decline to 0 by 2100, or before. Or, at the very least, acknowledge the absence of such a scenario as a potential weakness in the paper.

Response:

As we have already argued in our response to previous, similar concerns of Reviewer #2, we do not consider a scientifically valid approach what the Reviewer proposed, namely the assumption that the global economy will be truly net-zero in terms of CO₂ or GHG emissions and therefore that CO₂ emissions can simply be set to zero in an analysis like ours. Despite all net-zero goals and despite all positive developments in certain regions like Europe, we observe globally (i) continuously increasing GHG emissions, (ii) development of new fossil fuel reserves, and (iii) still massive expansion of fossil power generation. Carbon dioxide removal is still in its infancy and large-scale implementation remains to be proven.

However, as no one knows the future, we are happy to add the absence of such a net-zero scenario, in which electricity generation would be associated with zero GHG emissions also

from an LCA perspective, as an assumption in our analysis. We have added a corresponding sentence in the “Prospective life-cycle inventory database” section of the “Methods” part.

Comment:

Third, I agree with the authors' statement that aviation produces more climate impacts than other forms of travel. Clearly this is relevant when examining substitution effects (these are substitution effects, not rebound effects). But their article is examining the consequences of eliminating those climate impacts. And so at least in terms of economic issues, one needs to examine the economics. The simple fact is that aviation is the least expensive way to travel distances over a couple hundred kilometres, because of the very low infrastructure required. To make aviation climate neutral, fundamentally, will take a lot of infrastructure. So ultimately the question is whether one would need more infrastructure (at higher investment costs, and land use, and all that) with flying climate neutral, or travelling another way that is also climate neutral. This is why the conclusions reached in the final paragraph of the main manuscript (lines 355 - 364) need to be crafted carefully. From a physical standpoint, reducing aviation demand is attractive; we don't know if it is good until we compare the infrastructure effects that come with substitution. We also don't know if reducing aviation demand reduces the environmental and economic consequences, because (a) we are comparing two worlds that are both climate neutral, and (b) we haven't examined the environmental and economic consequences of the substitution that is likely to occur.

Response:

We do think that these conclusions (in the lines referred to by the Reviewer) are crafted carefully. We state that “...Thus, from a physical standpoint, reducing air-traffic demand is a good short- to mid-term solution. It drastically reduces the scale of the environmental and economic effort **needed to limit the impact of aviation on the climate.**”

On purpose, we limit the scope of our analysis to the aviation sector only and do not analyze any substitution effects in our overall mobility system. And we do think that such sectoral evaluations are valuable contributions to ongoing research. On top of that, one could even argue that the very minor reduction of air traffic, i.e., of less than 1% per year in our reduction scenarios, could be achieved by reducing mobility demand as such without substitution; such interesting consideration would lead to another discussion though, and we deliberately leave it out of the scope of our work.

Comment:

Finally, the sentence in the abstract containing "climate-neutral aviation will fly only if air traffic is reduced" will, simply put and from a scientific perspective, not fly. The conclusions simply do not support it. The paper finds that huge amounts of infrastructure are required. Yes. Will society be willing to build this infrastructure? Nobody knows. We build huge infrastructure all the time in order to achieve things we want to achieve. I suspect that if people really want to fly, and governments are clear that flying needs to be climate neutral, then the infrastructure will be built, and the cost of building it will be reflected in airline ticket prices. That's how the economy works. I tend to think this is the more likely outcome than governments' somehow restricting people's ability to fly, taking into account political, social, and economic factors.

Response:

We agree that reducing air traffic is not the only option for climate-neutral aviation. There are indeed alternatives. Therefore, we removed the adverb “only”. The abstract now ends with the following sentence:

“Here, we demonstrate that a European climate-neutral aviation will fly if air traffic is reduced to limit the scale of the climate impacts to mitigate.”